# Generalizable and Robust Spectral Method for Multi-view Representation Learning

**Amitai Yacobi**                                                       *amitaiyacobi@gmail.com*
*Department of Computer Science*
*Bar-Ilan University*

**Ofir Lindenbaum**                                                  *ofir.lindenbaum@biu.ac.il*
*Faculty of Engineering*
*Bar-Ilan University*

**Uri Shaham**                                                           *uri.shaham@biu.ac.il*
*Department of Computer Science*
*Bar-Ilan University*

**Reviewed on OpenReview:** *https://openreview.net/forum?id=X6IYO4Akw1*

## Abstract

Multi-view representation learning (MvRL) has garnered substantial attention in recent years, driven by the increasing demand for applications that can effectively process and analyze data from multiple sources. In this context, graph Laplacian-based MvRL methods have demonstrated remarkable success in representing multi-view data. However, these methods often struggle with generalization to new data and face challenges with scalability. Moreover, in many practical scenarios, multi-view data is contaminated by noise or outliers. In such cases, modern deep-learning-based MvRL approaches that rely on alignment or contrastive objectives present degraded performance in downstream tasks, as they may impose incorrect consistency between clear and corrupted data sources. We introduce *SpecRaGE*, a novel fusion-based framework that integrates the strengths of graph Laplacian methods with the power of deep learning to overcome these challenges. SpecRage uses neural networks to learn parametric mapping that approximates a joint diagonalization of graph Laplacians. This solution bypasses the need for alignment while enabling generalizable and scalable learning of informative and meaningful representations. Moreover, it incorporates a fusion module that dynamically adapts to data quality, ensuring robustness against outliers and noisy views. Our extensive experiments demonstrate that SpecRaGE outperforms state-of-the-art methods, particularly in scenarios with data contamination, paving the way for more reliable and efficient multi-view learning. Our code is available at: `https://github.com/shaham-lab/SpecRaGE`.

## 1 Introduction

Multi-view representation learning (MvRL) has become a crucial paradigm in recent years. Its primary objective is to integrate data from multiple sources into a unified representation, which can be used for various tasks such as clustering and classification. The demand for MvRL methods has surged as more applications require analyzing objects or phenomena from diverse perspectives. For instance, streaming platforms rely on the fusion of visual, audio, and textual features to enhance content recommendations, while healthcare systems combine genetic data, imaging, and clinical records to provide a comprehensive view of patient health.

Graph Laplacian methods have demonstrated notable effectiveness in representing both single-view and multi-view data (Ng et al., 2001; Belkin & Niyogi, 2003; Coifman & Lafon, 2006a; Cai et al., 2011; Kumar

et al., 2011; Eynard et al., 2012; 2015). This success largely stems from the ability of Laplacian eigenvectors to preserve similarity and capture the underlying cluster structure of the data (Ng et al., 2001; Belkin & Niyogi, 2003). Moreover, by focusing on the eigenvectors corresponding to the smallest eigenvalues, one can uncover the intrinsic low-dimensional manifold structure, as demonstrated in *Laplacian eigenmaps* (Belkin & Niyogi, 2003).

However, in the multi-view setting, traditional graph Laplacian-based methods face two major limitations: *Generalizability* - the ability to map new test points into the representation space after the training set has been processed (i.e., out-of-sample extension); *Scalability* – the capability to process large datasets within a practical time frame efficiently. Current graph Laplacian-based MvRL approaches often fail to address these two challenges (see discussion in Section 2). These limitations in generalizability and scalability hinder the full potential of graph Laplacians for representing multi-view data in real-world applications.

Another critical challenge in the multi-view setting is handling noise or outliers, which are pervasive in real-world applications. For example, in autonomous driving systems that integrate data from multiple sensors, adverse weather conditions can lead to unreliable or noisy data from certain sensors. Most modern deep learning-based MvRL approaches (Andrew et al., 2013; Huang et al., 2019; 2021; Federici et al., 2020; Trosten et al., 2021; Xu et al., 2022; Trosten et al., 2023; Wang et al., 2023; Yan et al., 2023) rely on alignment or contrastive objectives to enforce consistency between view-specific representations. These methods assume that all views of the same sample are of similar quality, making consistency enforcement reasonable under ideal conditions. However, this assumption breaks down in the presence of contaminated data, where enforcing consistency between clear and degraded views can lead to erroneous representations. These limitations underscore the urgent need for robust MvRL methods capable of handling real-world data imperfections.

To address the challenges of *Generalizability* and *Scalability* in graph Laplacian-based methods, as well as ensuring *Robustness* in the presence of contaminated views, we propose *SpecRaGE*, a novel fusion-based MvRL framework that combines the strengths of graph Laplacians with the power of deep learning. At its core, SpecRaGE provides a deep-learning solution to the approximate joint diagonalization of Laplacians problem, by extending *SpectralNet* (Shaham et al., 2018) to multi-view settings. The use of joint diagonalization is our key strategy for avoiding the need for alignment between views (see Section 4.1 for further discussion). SpecRaGE is inherently scalable, as it is trained on mini-batches in a stochastic manner, allowing it to efficiently process large datasets. Furthermore, the resulting model provides a parametric map that approximates the leading joint eigenvectors of the multi-view graph Laplacians. This parametric map enables efficient application to new data, addressing the generalizability challenge of traditional graph Laplacian methods.

Moreover, SpecRaGE incorporates a flexible fusion technique that overcomes the rigid limitations of traditional alignment-based methods when dealing with contaminated multi-view data. Specifically, SpecRaGE introduces a fusion mechanism, that generates sample-specific weight vectors, allowing the model to dynamically down-weight anomalous or noisy views.

Our extensive experiments demonstrate that SpecRaGE not only achieves state-of-the-art performance on standard multi-view benchmarks but also significantly outperforms existing methods when dealing with outliers and noisy views.

The main contributions of this work are: (1) We introduce a generalizable and scalable, graph Laplacian-based MvRL framework that extends the power of spectral methods to large-scale multi-view data. (2) We propose an adaptive fusion module that dynamically weights different views, providing robust performance in the face of data contamination. (3) We present extensive experimental results demonstrating SpecRaGE's superior performance across various benchmarks, particularly in scenarios involving outliers and noisy views.

## 2 Related Work

**Graph Laplacian-based MvRL Methods.** Various graph Laplacian-based methods (also known as multi-view spectral representation learning methods) have been proposed to extract compact and informative representations from multi-view data (Cai et al., 2011; Kumar et al., 2011; Eynard et al., 2012; 2015; Li et al., 2015; Lindenbaum et al., 2020; Yang et al., 2023). These approaches typically aim to learn a fused representation based on multiple graph Laplacians, one for each view. While these methods produce meaningful

representations, they often face challenges with generalizability (out-of-sample extension), requiring the recomputation of graph Laplacians to embed new, unseen data into the fused representation space. Some approaches address this issue through out-of-sample extension methods, such as the Nystrom extension (Nyström, 1930) or Geometric Harmonics (Coifman & Lafon, 2006b). However, these techniques were originally developed for single-view data and provide only local extensions, functioning effectively only near existing training points. Additionally, they are computationally intensive and memory-demanding, as they require calculating distances between each new test point and all training points. Furthermore, many graph Laplacian-based MvRL approaches face significant scalability challenges due to their computational complexity and memory requirements. These methods typically rely on iterative optimization procedures that demand substantial computations and storage for large Laplacians (Cai et al., 2011; Kumar et al., 2011; Eynard et al., 2012; 2015; Zhan et al., 2017; 2018). To address scalability, some approaches, such as (Li et al., 2015; Tao et al., 2024), introduce large-scale techniques for multi-view spectral clustering. (Li et al., 2015) improves efficiency by constructing a bipartite graph to approximate the full affinity matrix, reducing the computational complexity of spectral clustering. Tao et al. (2024) enhances scalability by leveraging tensor factorization to decompose multi-view data into a compact shared latent space, avoiding the need for explicit pairwise similarity computations. However, the method presented in (Tao et al., 2024) still requires quadratic time complexity, and both methods are specifically designed for clustering rather than learning representations. As a result, they cannot directly map new, unseen points into the learned representation space. These limitations in generalizability and scalability hinder the practical use of graph Laplacians for multi-view representation learning in real-world applications.

**Deep-learning based MvRL Methods.** Modern deep-learning-based MvRL methods attempt to design a loss function that is useful for extracting meaningful representations from multi-view data. One category of methods includes deep extensions of the Canonical Correlation Analysis (CCA) algorithm, such as DCCA, DCCAE, and $\ell_0$-DCCA (Hotelling, 1936; Andrew et al., 2013; Wang et al., 2015; Lindenbaum et al., 2021). These methods utilize deep networks to learn a non-linear mapping for two views, maximizing their correlations. Another set of algorithms relies on information-theory-based metrics (Federici et al., 2020; Lin et al., 2022), aiming to maximize the mutual information between views while minimizing redundant information unique to each view. Contrastive learning methods (Trosten et al., 2021; Xu et al., 2022; Yan et al., 2023; Wang et al., 2023; Guo et al., 2024) represent another group of deep learning approaches, utilizing a contrastive alignment objective to achieve view-specific alignment. Deep learning techniques have also been applied to address the multi-view spectral clustering problem (Huang et al., 2019; 2021). These methods are closely related to our work, as they also extend SpectralNet to multi-view settings by incorporating alignment objectives between the spectral embeddings from each view. Despite their promising performance, these approaches rely on some form of alignment between the view-specific representations, making them vulnerable to data contamination. Their underlying assumptions regarding data quality and view consistency may struggle in the presence of noise, outliers, or asymmetric corruption across views, as demonstrated in Section 5.3.

**Robustness in Multi-view Learning.** Recent works have explored various approaches to handling corruption in multi-view settings. Geng et al. (2021); Zhang et al. (2023) introduce an uncertainty estimation mechanism to model reliability across views through probabilistic modeling. However, these methods are specifically designed for Gaussian noise, making strong assumptions about the nature of uncertainty that may not hold in real-world scenarios with complex corruption patterns. Xu et al. (2020) approaches robustness through modal regression and low-rank matrix recovery, formulating multi-view learning as a structured matrix recovery problem. While this method can handle outliers, it requires solving complex optimization problems involving alternate minimization with multiple terms. Other approaches address different aspects of multi-view noise that are less relevant to our work. Sun et al. (2024) focuses on scenarios where sample correspondences between views are noisy, meaning that some instances are incorrectly paired across views. While their probabilistic model corrects such correspondences during clustering, this method addresses a fundamentally different problem from our goal of handling corrupted observations within properly aligned views. Similarly, Xu et al. (2024) primarily deals with label noise rather than feature-level corruption, introducing a trustworthiness mechanism to reweight samples based on their likelihood of having incorrect labels. This focus on supervision noise makes it orthogonal to our problem of handling corruption in the views themselves.

## 3 Preliminaries

### 3.1 Graph Laplacian and SpectralNet

**Graph Laplacian.** Given a dataset of $n$ points $x_1, x_2, \ldots, x_n$, an affinity matrix $W$ is an $n \times n$ symmetric matrix with non-negative entries, where $W_{i,j}$ represents the similarity between $x_i$ and $x_j$. The unnormalized graph Laplacian is defined as $L = D - W$ where $D$ is a diagonal matrix in which the element $D_{i,i} = \sum_{j=1}^{n} W_{i,j}$ correspond to the degrees of the points $x_i$, $i = 1, \ldots, n$. The eigenvectors corresponding to the smallest eigenvalues of the Laplacian provide valuable low-dimensional representations, capturing structural information like relationships and similarities between data points. These eigenvectors are widely used in applications such as spectral clustering (Ng et al., 2001), dimensionality reduction (Belkin & Niyogi, 2003), graph partitioning (Karypis & Kumar, 1998), and image segmentation (Shi & Malik, 2000; Melas-Kyriazi et al., 2022).

**SpectralNet.** *SpectralNet* (Shaham et al., 2018) is a deep-learning model that effectively maps single-view data to the approximate eigenvectors of its Laplacian. This enables the performance of spectral clustering on huge single-view datasets since the loss is amenable to parallelized training. Furthermore, the model can be easily and accurately used to generalize the representation to unseen test data. To learn the eigenvectors, *SpectralNet* minimizes the following Rayleigh-quotient loss: $\mathrm{Tr}\left(Y^\top L Y\right)$ s.t. $Y^\top Y = I$, where $Y \in \mathbb{R}^{n \times k}$ is the network's output and $L$ is the graph Laplacian.

### 3.2 Joint Diagonalization

**Definition 1.** *A set of diagonalizable matrices $A^{(1)}, A^{(2)}, \ldots, A^{(V)}$ is said to be simultaneously diagonalizable if there exists a single invertible matrix $U$ such that for all $1 \leq v \leq V$, $U^{-1} A^{(v)} U$ is a diagonal matrix.*

Intuitively, *joint diagonalization* (or *simultaneous diagonalization*) seeks to find a common basis by which all matrices could be represented in a diagonal form.

However, exact joint diagonalization can be achieved if and only if $A^{(1)}, A^{(2)}, \ldots, A^{(V)}$ commute. Nevertheless, when commutativity cannot be guaranteed, optimizing a joint diagonality criterion and approximating the solution remains possible. This defines the *approximate joint diagonalization* problem.

**Approximate Joint Diagonalization of Laplacians.** In terms of graph Laplacians, for a set of graph Laplacians $L^{(1)}, L^{(2)}, \ldots, L^{(V)}$ the objective of *joint diagonalization* is to find a set of orthogonal eigenvectors $U$ such that for each Laplacian $L^{(v)}$, the matrix $U^\top L^{(v)} U$ is diagonal and contains the eigenvalues of $L^{(v)}$ on the diagonal. It has been demonstrated in (Eynard et al., 2012; 2015), that it is possible to find an approximate joint diagonalization of Laplacians by solving the following objective:

$$\min_{U \in \mathbb{R}^{n \times k}} \mathrm{Tr}\left(U^\top \bar{L} U\right), \quad \text{s.t.} \quad U^\top U = I, \tag{1}$$

where $\bar{L}$ represents a form of average of the graph Laplacians. This average can be computed, for example, as a weighted arithmetic mean: $\bar{L} = \sum_{v=1}^{V} \alpha^{(v)} L^{(v)}$ where $\alpha^{(v)}$ represents the contribution of the $v$-th view. In Section 4.5, we provide a more detailed discussion of this objective and its connection to approximate joint diagonalization.

## 4 SpecRaGE

**Problem Statement.** Let $X = \{\mathcal{X}^{(v)} \in \mathbb{R}^{n \times d_v}\}_{v=1}^{V}$ be a multi-view dataset where $n$ is the number of samples, $V$ is the number of views, and $d_v$ denotes the dimensionality of samples within the $v$-th view. MvRL aims to leverage the multi-view information to learn a high-quality unified representation that facilitates downstream tasks such as clustering, manifold learning, or classification.

In this section, we introduce SpecRaGE, our MvRL framework designed to tackle the challenges of generalizability and scalability in graph Laplacian methods, and robustness to contaminated views. We begin by outlining the rationale behind our method. Next, we detail how SpecRaGE efficiently learns the approximate

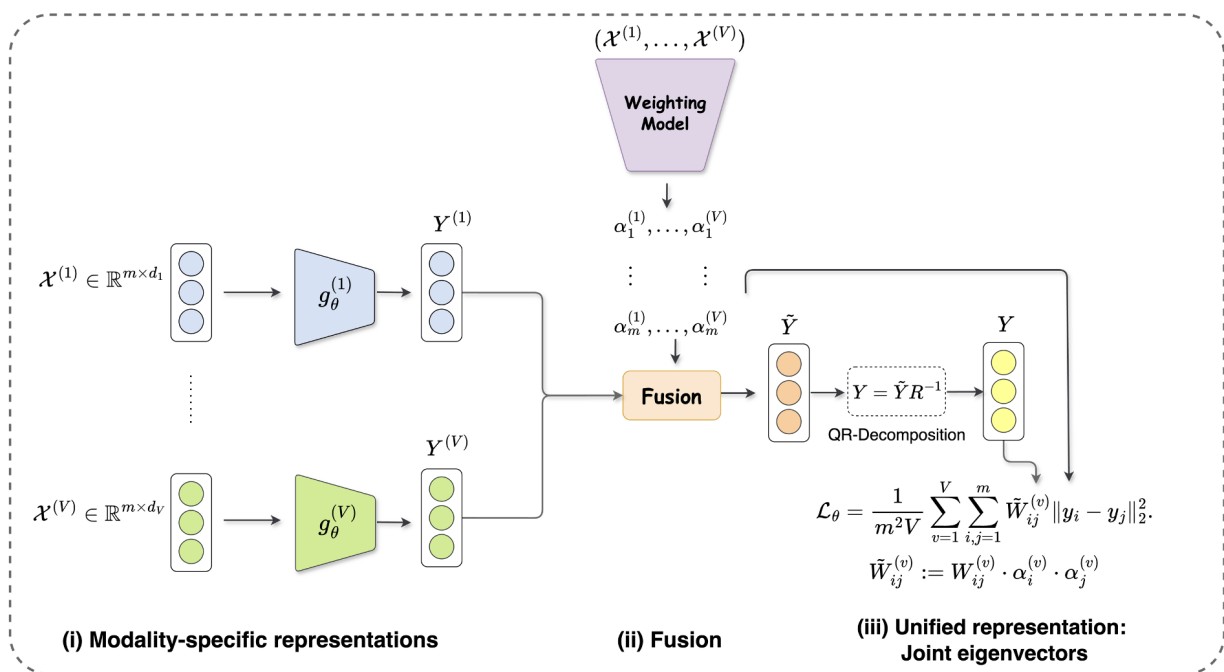

Figure 1: SpecRaGE architecture: Given an input of $V$ batches, each containing $m$ samples, $V$ view-specific representations are generated by $V$ corresponding neural networks. Subsequently, the concatenated input is passed through the fusion model, which computes the weights for performing the weighted sum fusion. The resulting fused representation undergoes QR decomposition to enforce orthogonality. Finally, the loss function in Eq. 4 is computed, and the weights of all networks are updated via the gradients.

joint eigenvectors, facilitating generalization to new data. Finally, we explore the fusion module, which enhances robustness in the presence of contaminated views. The overall framework of our method is illustrated in Fig. 1.

## 4.1 Rationale

To effectively harness the strengths of graph Laplacians while ensuring robustness to data contamination, we aim to: (1) adopt a less contamination-sensitive strategy that eliminates the need for alignment objectives, (2) implement a dynamic weighting mechanism to reduce the influence of low-quality views, and (3) provide a scalable and generalizable graph Laplacian solution.

SpecRaGE addresses the first requirement through joint diagonalization of Laplacians, which fuses information from all views into a unified representation that approximates the joint eigenvectors. This fusion process involves averaging Laplacians across views, which naturally reduces the impact of corrupted data, as noise in one view is typically independent of others. This makes SpecRaGE less sensitive to contamination because the impact of noisy views is diluted when averaged with cleaner views. In contrast, alignment-based approaches directly enforce consistency between representations of all views, which can lead to clean views being compromised when forced to align with contaminated ones.

To meet the second requirement, SpecRaGE incorporates a dynamic fusion module that adjusts the weighting of views based on their quality, effectively reducing the influence of noisy or outlier views in a sample-based manner. This module prioritizes cleaner views while down-weighting those affected by noise or corruption, ensuring robust and accurate fused representations.

Overall, SpecRaGE is a parametric (hence generalizable) map, trained in a stochastic fashion (hence scalable), and therefore meets the third requirement as well.

## 4.2 Generalizable and Scalable Approximation of Joint Eigenvectors

Let $F_\theta : \mathbb{R}^{d_1} \times \mathbb{R}^{d_2} \times \cdots \times \mathbb{R}^{d_V} \to \mathbb{R}^k$ be a parametric mapping (i.e., a neural network model) that transforms a multi-view input into the corresponding coordinates in a fused representation. That is, for a multi-view input $\hat{x}_i = \left( x_i^{(1)}, x_i^{(2)}, \ldots, x_i^{(V)} \right)$, $F_\theta$ produces $y_i = F_\theta(\hat{x}_i)$ where $y_i \in \mathbb{R}^k$ represents coordinates in the fused representation.

Given a batch of $m$ multi-view data points, let $W^{(v)}$ denote the $m \times m$ affinity matrix for the $v$-th view, where $W_{i,j}^{(v)}$ represents an affinity measure between $x_i^{(v)}$ and $x_j^{(v)}$. We propose the following loss:

$$\mathcal{L}_\theta = \frac{1}{m^2 V} \sum_{v=1}^{V} \sum_{i,j=1}^{m} W_{i,j}^{(v)} \|y_i - y_j\|_2^2, \tag{2}$$

This loss encourages points with high affinity (as measured by the affinity matrices) to be close in the fused representation space. However, it is evident that this loss can be minimized trivially by mapping all points to the same output, i.e., $F_\theta(\hat{x}_i) = y_0$ for all $i$. To avoid this trivial solution, an orthogonality constraint is added to the outputs:

$$Y^\top Y = I_{k \times k},$$

where $Y$ is an $m \times k$ matrix of outputs, with its $i$-th row corresponding to $y_i$. To satisfy the orthogonality constraint, we follow the same technique from (Shaham et al., 2018) and construct an orthogonalization layer that computes the QR decomposition of the network output and returns the orthogonal $Q$ matrix. More specifically, let $\tilde{Y}$ denote the $m \times k$ matrix obtained from the fusion step; the weights of this layer are defined to be the matrix $R^{-1}$ from the QR decomposition of $\tilde{Y}$. The final orthogonal output is then $Y = \tilde{Y} R^{-1}$. For further details about the training process with the orthogonalization layer, see Appendix F.

With some mathematical transitions (see Appendix C), the loss in Eq. 2 can be written in the following Rayleigh quotient form:

$$\mathcal{L}_\theta = \frac{2}{m^2 V} \operatorname{Tr} \left( Y^\top \sum_{v=1}^{V} L^{(v)} Y \right), \tag{3}$$

where $L^{(v)}$ is the $m \times m$ graph Laplacian of the $v$-th view. One can observe that this loss is exactly the arithmetic mean version of the objective in Eq. 1. In Section 4.5 we further explore the connection between this loss function and the approximate joint diagonalization of Laplacians problem.

The choice of affinity measure plays a crucial role in determining the quality of the generated representations. An appropriate affinity measure can enhance the ability of the model to capture the underlying relationships within the data, while an inadequate one may lead to poor representation quality and misinterpretation of the data structure. In Section 4.4, we provide a discussion of the technique we employed to construct the affinity matrices.

**Generalizability and Scalability.** Once the framework is trained, it provides a mapping function $F_\theta$ that transforms each multi-view sample directly into its coordinates in the final unified representation, facilitating efficient generalization for new samples from the same distribution. Notably, all our experiments were conducted using test sets, which illustrates the method's generalizability. Additionally, the stochastic mini-batch training in SpecRaGE avoids computing the full Laplacian eigenvectors, enabling scalability for large datasets. For example, SpecRaGE processed the 1-million-sample InfiniteMNIST dataset (see Section 5.1) in about 15 minutes on a MacBook with M1 CPU, whereas traditional graph Laplacian methods faced out-of-memory errors or much longer runtimes. The overall running time complexity of SpecRaGE is $O(n(k^2 + mV))$, where $k$ is the output dimension, $V$ is the number of views, and $m$ is the batch size. Since $k$, $V$, and $m$ are typically much smaller than $n$ and independent of $n$, our method exhibits near-linear time complexity. In Appendix D.3, we compare the runtime of our method against several existing graph Laplacian-based approaches. In Appendix E, we present the full time complexity analysis.

### 4.3 Robust Weighting Fusion Model

**View-specific Representations.** As described above, the loss functions in Eq. 2 and Eq. 3 operate on a fused representation $Y \in \mathbb{R}^{m \times k}$, derived from a mini-batch of $m$ multi-view samples. To construct this fused representation, we first need to generate intermediate representations for each individual view. The primary goal of these representations is to embed numeric and categorical features into a common space and ensure that all views are of the same size. To extract the view-specific representations, we introduce an individual neural network, $g_\theta^{(v)} : \mathbb{R}^{d_v} \to \mathbb{R}^k$ (e.g., an MLP), for each view. Specifically, given a multi-view input $\hat{x}_i = \left( x_i^{(1)}, x_i^{(2)}, \ldots, x_i^{(V)} \right)$, the intermediate representations are $y_i^{(1)}, y_i^{(2)}, \ldots, y_i^{(V)}$, where $y_i^{(v)} = g_\theta^{(v)} \left( x_i^{(v)} \right)$. For a batch of $m$ multi-view samples, we obtain $V$ matrices $Y^{(1)}, Y^{(2)}, \ldots, Y^{(V)}$, where $Y^{(v)}$ is an $m \times k$ matrix of the outputs, and its $i$-th row corresponds to $y_i^{(v)}$.

Merging the view-specific representations $Y^{(1)}, Y^{(2)}, \ldots, Y^{(V)}$ into a unified representation $Y$ presents a significant challenge. A key difficulty in this process arises from the potential presence of contaminated samples in some views, such as outliers or noisy data. In such cases, it is desirable to give less weight to the low-quality views in order to obtain a more accurate representation. Specifically, we need an approach that can evaluate the quality of each view directly from the data and determine its degree of contribution accordingly.

**Dynamic Weighting Model for Fusion.** To dynamically weight views based on their quality, we introduce another neural network model (specifically, an MLP) that takes a multi-view sample $\hat{x}_i$ as input and predicts a weights vector $\alpha_i \in \Delta_V$, where $\Delta_V = \{\alpha \in \mathbb{R}^V : \alpha^{(v)} \geq 0, \text{for } v = 1, \ldots V, \ \sum_{v=1}^V \alpha^{(v)} = 1\}$ is the $V$ dimensional probability simplex. Each entry of $\alpha_i$ indicates the quality of the corresponding view for instance $\hat{x}_i$. The process works as follows: First, the view-specific representations $y_i^{(1)}, y_i^{(2)}, \ldots, y_i^{(V)}$ are obtained from the multi-view sample $\hat{x}_i$. Then, $\hat{x}_i$ is concatenated and passed through the weighting model, generating a weights vector $\alpha_i$. Finally, the fused representation is computed as $y_i = \sum_{v=1}^V \alpha_i^{(v)} \cdot y_i^{(v)}$.

To provide meaningful feedback to the weighting model, we integrate the view-specific contributions directly into the similarity matrices in Eq. 2. Specifically, for a batch of size $m$, weight vectors $\alpha_1, \alpha_2, \ldots, \alpha_m$ are first generated using the weighting model, where $\alpha_i^{(v)}$ represents the reliability of sample $i$ in the $v$-th view. These weights are incorporated by modifying each entry in the similarity matrix: $\tilde{W}_{ij}^{(v)} := W_{ij}^{(v)} \cdot \alpha_i^{(v)} \cdot \alpha_j^{(v)}$. This multiplication adjusts the strength of pairwise relationships based on the reliability of both samples involved. The resulting loss function is:

$$\mathcal{L}_\theta = \frac{1}{m^2 V} \sum_{v=1}^V \sum_{i,j=1}^m \tilde{W}_{ij}^{(v)} \|y_i - y_j\|_2^2. \tag{4}$$

This formulation allows the weighting model to dynamically adjust the contribution of each view. By multiplying $W_{ij}^{(v)}$ with $\alpha_i^{(v)}$ and $\alpha_j^{(v)}$, the model effectively reduces the influence of noisy or corrupted samples, ensuring that their similarities to other samples are down-weighted while preserving the structure of reliable ones. If the weighting model assigned high weights to contaminated views, the loss would increase, as noisy samples would distort the similarity structure in the corresponding $W^{(v)}$ and lead to suboptimal representations, reinforcing the need for correctly learned weights. A complete algorithm summarizing our method can be found in Alg. 1.

To demonstrate the capability of our fusion mechanism to identify observations of low quality and downscale their importance in the overall objective, we conducted an experiment with the scikit-learn 2D Blobs dataset[1]. We created two views, where the second view is a random rotation of the first, and randomly injected outliers independently into each view, comprising 20% of the samples. After training SpecRaGE on this dataset, we analyzed the weight distributions of clean and contaminated samples in a test set. As illustrated in Fig. 2, our method effectively distinguishes between clean and contaminated samples through their assigned weights. Clean samples consistently receive weights centered around 0.5, indicating balanced and reliable information.

---

[1]https://scikit-learn.org/stable/modules/generated/sklearn.datasets.make_blobs.html

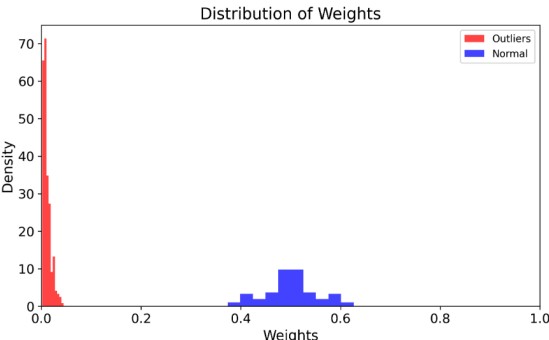

Figure 2: Distribution of weights assigned by our weighting model on the scikit-learn 2D Blobs dataset. Clean samples (blue) receive moderate weights centered around 0.5, while outliers (red), are assigned very low weights, effectively reducing their influence on the learning process.

In contrast, outlier samples, are assigned significantly lower weights (close to 0), effectively reducing their influence on the learned representations. This clear separation in weight distributions demonstrates the model's ability to automatically identify and downweight potentially corrupted samples. Additionally, Section 5.3 demonstrates that this approach not only achieves state-of-the-art performance on standard multi-view benchmarks but also significantly outperforms existing methods in handling outliers and noisy views.

### 4.4 Self-supervised Affinity Learning.

To perform approximate joint diagonalization of Laplacians, it is necessary to know how to construct an affinity matrix for each view. A widely used approach is to use a Gaussian kernel with a specific scale parameter $\sigma > 0$. For instance, for the $v$-th view, the affinity matrix is defined as follows:

$$W_{i,j}^{(v)} = \begin{cases} \exp\left(-\frac{\|x_i^{(v)} - x_j^{(v)}\|^2}{2\sigma^2}\right), & x_j^{(v)} \text{ is one of the } l \text{ nearest neighbors of } x_i^{(v)}, \\ 0, & \text{otherwise.} \end{cases} \tag{5}$$

However, Euclidean distance may offer a limited measure of similarity, particularly for high-dimensional data (Beyer et al., 1999; Aggarwal et al., 2001). Therefore, instead of directly computing the Euclidean distance between $x_i$ and $x_j$, we choose to train a SiameseNet (Koch et al., 2015) for each view $v$, denoted as $h_{\theta_{\text{siamese}}}^{(v)}$. This replaces the Euclidean distance $\|x_i^{(v)} - x_j^{(v)}\|^2$ in Eq. 5 with $\|z_i^{(v)} - z_j^{(v)}\|^2$, where $z_i^{(v)} = h_{\theta_{\text{siamese}}}^{(v)}\left(x_i^{(v)}\right)$.

Training a SiameseNet typically requires positive and negative pairs. When labels are provided, pairs from the same class are considered positive, while those from different classes are negative. In our unsupervised setup, we determine positive pairs based on small Euclidean distances between points $x_i^{(v)}$ and $x_j^{(v)}$, and negative pairs otherwise. Specifically, positive pairs are formed from the $l$ nearest neighbors of each point, while negative pairs are generated from points that are not included in the nearest neighbors set. The parameters $l$ and $\sigma$ are hyper-parameters, discussed in Appendix F.

The training of the SiameseNets serves as a pre-processing step. Once the SiameseNets are trained, we use them to construct batch affinity matrices for each view during the training. As in (Shaham et al., 2018), we empirically found that using SiameseNet for the affinities improves the quality of the representations.

### 4.5 Theoretical Analysis

In this section, we provide additional theoretical insight into the relationship between our loss function in Eq. 3 and the approximate joint diagonalization problem. We begin by establishing a fundamental property of jointly diagonalizable matrices:

**Proposition 1.** *Let $L^{(1)}, L^{(2)}, \ldots, L^{(V)}$ be $n \times n$ real, symmetric, and pairwise commuting matrices. Let $\bar{L} = \sum_{v=1}^{V} L^{(v)}$ be a sum of these matrices. Then, the joint eigenvectors of $L^{(1)}, \ldots, L^{(V)}$ are also the eigenvectors of $\bar{L}$.*

The proof of this proposition can be found in Appendix B.

Due to the uniqueness of the eigendecomposition of $\bar{L}$, these joint eigenvectors, are the only eigenvectors of $\bar{L}$ (up to sign). Therefore, the joint eigenvectors of $L^{(1)}, L^{(2)}, \ldots, L^{(V)}$ can be recovered by analyzing the eigendecomposition of their sum, $\bar{L}$. Building on this result, we can now characterize the optimal solution for our loss functions:

**Corollary 1.** *When joint eigenvectors exist, the minimizer of the loss functions in Eq. 3 consists of the joint eigenvectors (or an orthogonal rotation of them) of $L^{(1)}, L^{(2)}, \ldots, L^{(V)}$ corresponding to the k smallest eigenvalues of $\bar{L} = \sum_{v=1}^{V} L^{(v)}$.*

This corollary follows from both Proposition 1 and classical Rayleigh-quotient optimization. Classical Rayleigh-quotient optimization tells us that minimizing Eq. 3 is equivalent to finding the eigenvectors of $\bar{L}$ corresponding to its smallest eigenvalues. Proposition 1 further establishes that these eigenvectors are the joint eigenvectors of the matrices $L^{(1)}, L^{(2)}, ..., L^{(V)}$. Consequently, the minimizers of the loss functions are intrinsically tied to the joint eigenvectors, thereby highlighting the connection between our optimization problem and the joint diagonalization of these matrices.

In practical scenarios, the graph Laplacians $L^{(1)}, L^{(2)}, ..., L^{(V)}$ rarely satisfy the commutativity condition perfectly due to contaminations and inherent differences between modalities. In such cases, our loss functions serve to find approximate joint eigenvectors, providing an approach to handling real-world data where exact joint diagonalization may not be possible. Appendix D.4, shows that our method effectively approximates the eigenvectors of $\bar{L}$ and performs approximate joint diagonalization.

## 5 Experiments

### 5.1 Experimental Settings

**Datasets.** We assess the performance of SpecRaGE using five well-studied multi-view datasets. Our selection prioritizes datasets that exhibit diversity in the types and number of views, as well as the number of classes, as depicted in Table 3 in Appendix A. The datasets are listed as follows: (1) **BDGP** (Cai et al., 2012) contains 2500 images of Drosophila embryos divided into five categories with two extracted features. One view has 1750-dimensional visual features, and the other view has 79-dimensional textual features. (2) **Reuters** is a multilingual dataset comprising more than 11,000 articles across six categories and five languages: English, French, German, Italian, and Spanish. We used a subset of this dataset containing 18,758 samples for our analysis. (3) **Caltech20** is a subset of 2386 examples derived from the object recognition dataset (Fei-Fei et al., 2004), which comprises 20 classes viewed from six different perspectives. The dataset encompasses various features such as Gabor features, wavelet moments, CENTRIST features, histogram of oriented gradients, GIST features, and local binary patterns. (4) **Handwritten** (Asuncion & Newman, 2007) contains 2,000 digital images of handwritten numerals ranging from 0 to 9. This dataset employs two types of descriptors: a 240-dimensional pixel average within $2 \times 3$ windows, and a 216-dimensional profile correlations method, serving as two distinct viewpoints. (5) **InfiniteMNIST** is a large-scale variant of MNIST (LeCun et al., 1998), with 1 million samples. The first view contains the original images, while the second adds Gaussian noise ($\sigma = 0.2$) to images from the same class, as in (Trosten et al., 2023). Created similarly to affNIST[2], this dataset applies small random affine transformations to expand the original data. More details about the different datasets can be found in Appendix A.

**Baselines.** We compared the performance of SpecRaGE with seven multi-view representation learning methods including two classic deep methods (DCCA (Andrew et al., 2013) and DCCAE (Wang et al., 2015)) and five state-of-the-art methods (MvSCN (Huang et al., 2019), MIB (Federici et al., 2020), Multi-VAE (Xu

---

[2]https://www.cs.toronto.edu/~tijmen/affNIST/

Table 1: Clustering results of all methods on five datasets. The best result in each row is shown in bold and the second-best is underlined.

| Datasets | Metrics | DCCA | DCCAE | MvSCN | MIB | Multi-VAE | AECoKM | MetaViewer | SpecRaGE |
|---|---|---|---|---|---|---|---|---|---|
| BDGP | ACC | 75.8 ±1.31 | 80.1 ±1.50 | 81.9 ±4.33 | 86.9 ±0.75 | 53.8 ±6.91 | 76.6 ±5.33 | 90.4 ±0.43 | **97.6** ±0.53 |
| | NMI | 67.9 ±1.15 | 73.2 ±1.04 | 76.9 ±4.15 | 80.9 ±0.91 | 29.9 ±6.54 | 68.7 ±5.89 | 86.3 ±1.05 | **93.4** ±0.83 |
| | ARI | 49.2 ±1.31 | 55.1 ±1.02 | 70.3 ±5.42 | 74.1 ±1.82 | 28.9 ±4.92 | 54.7 ±6.34 | 89.3 ±1.21 | **94.1** ±1.29 |
| Reuters | ACC | 47.9 ±0.93 | 42.0 ±1.23 | 49.5 ±2.50 | 49.5 ±1.10 | 36.6 ±2.34 | 26.5 ±0.52 | 47.8 ±0.08 | **56.7** ±3.22 |
| | NMI | 26.6 ±1.32 | 20.3 ±1.13 | 27.6 ±1.32 | 28.4 ±1.81 | 21.1 ±2.20 | 16.4 ±1.54 | 24.2 ±0.06 | **38.3** ±2.45 |
| | ARI | 12.7 ±1.44 | 8.5 ±1.06 | 23.1 ±1.12 | 24.4 ±1.23 | 12.4 ±2.34 | 10.1 ±2.54 | 19.19 ±0.81 | **29.6** ±2.64 |
| Caltech20 | ACC | 39.7 ±1.10 | 36.1 ±1.50 | 42.1 ±2.59 | 36.1 ±1.23 | 32.4 ±2.25 | 42.2 ±3.65 | 45.1 ±1.33 | **50.1** ±2.61 |
| | NMI | 51.2 ±1.21 | 52.4 ±1.71 | 53.7 ±3.49 | 47.5 ±1.55 | 46.2 ±2.82 | 57.6 ±0.18 | 60.9 ±2.15 | **66.0** ±1.72 |
| | ARI | 23.5 ±2.23 | 25.3 ±1.14 | 29.7 ±3.26 | 23.2 ±1.04 | 24.8 ±1.79 | 32.2 ±1.20 | 35.0 ±1.24 | **40.1** ±3.49 |
| Handwritten | ACC | 58.3 ±2.10 | 63.8 ±1.21 | 68.3 ±4.86 | 67.5 ±2.95 | 74.5 ±3.34 | 66.4 ±2.34 | 86.3 ±2.51 | **91.9** ±3.51 |
| | NMI | 70.2 ±1.69 | 75.2 ±1.34 | 70.9 ±2.90 | 64.7 ±2.18 | 70.0 ±3.65 | 70.3 ±2.70 | 78.9 ±1.44 | **86.5** ±1.74 |
| | ARI | 52.2 ±1.33 | 60.8 ±1.87 | 59.6 ±4.25 | 48.9 ±2.56 | 60.5 ±3.15 | 61.4 ±2.70 | 72.3 ±2.94 | **83.0** ±1.98 |
| InfiniteMNIST | ACC | 95.6 ±1.09 | 95.1 ±1.21 | 99.1 ±0.28 | 68.6 ±3.95 | 98.1 ±0.50 | **99.3** ±0.20 | 80.0 ±0.22 | 99.1 ±0.27 |
| | NMI | 90.0 ±1.39 | 88.9 ±1.41 | 97.7 ±1.54 | 67.2 ±3.15 | 96.0 ±1.20 | **98.1** ±0.42 | 72.3 ±0.15 | 97.5 ±0.54 |
| | ARI | 90.1 ±1.33 | 89.3 ±1.01 | 97.9 ±0.48 | 66.9 ±3.56 | 96.4 ±1.05 | **98.5** ±0.20 | 65.2 ±0.24 | 97.9 ±0.21 |

et al., 2021), AECoKM (Trosten et al., 2023), and MetaViewer (Wang et al., 2023)). DCCA and DCCAE are deep extensions of classic correlation analysis. MvSCN learns spectral embeddings for each view while aligning view-specific representations with an MSE objective. MIB uses information theory to separate shared and view-specific information. Multi-VAE disentangles visual representations into view-common and view-peculiar variables. AECoKM combines autoencoders with contrastive learning for view alignment. MetaViewer learns to fuse representations by observing reconstruction from unified representations to specific views and employs contrastive learning in its improved version. To ensure fairness, we ran each algorithm ten times on the aforementioned datasets using the same backbones, recording the mean and standard deviation of their performance. For clustering, we employed $k$-means, while Support Vector Machines (SVM) were used for classification. For alignment-based methods, we concatenated the view-specific representations before applying clustering and classification. More details on hyper-parameters and training are in Appendix F.

**Evaluation metrics.** For clustering, we employed three widely used metrics: Normalized Mutual Information (NMI), Accuracy (ACC), and Adjusted Rand Index (ARI). For classification, we used Accuracy, Precision, and F-score. Higher values of these metrics indicate superior performance.

## 5.2 Evaluation of the Learned Representations

**Clustering Results.** In the clustering experiment, we aimed to evaluate SpecRaGE's ability to capture the underlying cluster structure of the data in comparison to the baseline methods. To achieve this, we applied the $k$-means algorithm to the final representations learned by each method. Table 1 summarizes the clustering results across the five datasets. SpecRaGE achieves top-performing results on most datasets and evaluation metrics, with particularly notable improvements on BDGP and Reuters. On these datasets, SpecRaGE outperforms the second-best method by more than 7%. The consistent clustering performance across different metrics (ACC, NMI, and ARI) suggests that SpecRaGE learns representations that preserve both global cluster structure and local neighborhood relationships. Only on InfiniteMNIST, where most methods already perform well due to the dataset's relative simplicity, does SpecRaGE perform comparably to the best baseline. In Figure 3 we provide an additional illustration of the efficacy of SpecRaGE in representing the data and capturing its inherent cluster structure.

**Classification Results.** In the classification experiment, we aimed to assess the quality of the learned representations by running an SVM on top of the final embeddings produced by each method. We chose to use SVM because it does not introduce any additional non-linear transformations to the feature spaces, ensuring that the comparison between different algorithms remains unbiased. SVM and linear classifiers are widely accepted as standard benchmarks for evaluating learned representations and embeddings (See, for example, (Chen et al., 2020; Federici et al., 2020; Bardes et al., 2021; Wang et al., 2023)).

Table 2: Classification results of all methods on five datasets. The best result in each row is shown in bold and the second-best is underlined.

| Datasets | Metrics | DCCA | DCCAE | MvSCN | MIB | Multi-VAE | AECoKM | MetaViewer | SpecRaGE |
|---|---|---|---|---|---|---|---|---|---|
| BDGP | ACC | 98.40 ±0.81 | 98.65 ±0.26 | 98.76 ±0.10 | 90.56 ±1.55 | 88.87 ±2.54 | 98.92 ±0.21 | 98.00 ±0.11 | **99.01** ±0.30 |
| | F-score | 98.40 ±0.82 | 98.50 ±0.21 | 98.76 ±0.10 | 90.10 ±0.60 | 88.87 ±2.54 | 98.92 ±0.20 | 98.02 ±0.11 | **99.00** ±0.23 |
| | Precision | 98.42 ±0.61 | 98.63 ±0.30 | 98.02 ±0.15 | 89.92 ±1.32 | 89.07 ±2.43 | 99.01 ±0.20 | 98.59 ±0.10 | **99.01** ±0.20 |
| Reuters | ACC | 74.40 ±0.92 | 74.10 ±0.87 | 75.52 ±0.12 | 71.96 ±1.02 | 62.06 ±3.40 | 68.01 ±0.85 | 59.17 ±0.10 | **76.61** ±1.70 |
| | F-score | 74.50 ±1.10 | 74.21 ±0.90 | 75.51 ±0.12 | 70.78 ±0.91 | 59.60 ±3.95 | 65.21 ±0.81 | 51.19 ±0.09 | **76.60** ±1.77 |
| | Precision | 74.72 ±0.92 | 74.35 ±1.01 | 75.53 ±0.14 | 70.78 ±1.10 | 61.31 ±3.22 | 67.24 ±0.85 | 56.43 ±0.12 | **78.59** ±1.75 |
| Caltech20 | ACC | 72.60 ±0.51 | 72.58 ±0.64 | 86.54 ±0.35 | 73.52 ±2.41 | 87.73 ±0.63 | 93.38 ±1.07 | 92.16 ±0.05 | **95.42** ±0.92 |
| | F-score | 40.12 ±0.50 | 43.26 ±0.81 | 86.75 ±0.37 | 72.52 ±2.10 | 87.30 ±0.91 | 93.03 ±0.97 | 85.72 ±0.10 | **95.93** ±0.93 |
| | Precision | 46.30 ±0.44 | 60.66 ±0.69 | 85.32 ±0.33 | 73.25 ±1.93 | 88.90 ±1.60 | 93.69 ±1.26 | 90.68 ±0.15 | **95.44** ±0.90 |
| Handwritten | ACC | 88.25 ±0.91 | 90.01 ±0.45 | 96.21 ±0.21 | 96.01 ±1.04 | 95.36 ±0.95 | 96.91 ±0.65 | 97.75 ±0.21 | **97.80** ±1.27 |
| | F-score | 88.05 ±1.03 | 89.92 ±0.45 | 96.21 ±0.21 | 96.10 ±1.20 | 95.36 ±0.93 | 96.95 ±0.63 | 97.75 ±0.20 | **97.77** ±1.27 |
| | Precision | 89.20 ±0.85 | 90.48 ±0.56 | 96.24 ±0.21 | 96.03 ±0.90 | 95.38 ±0.92 | 96.91 ±0.34 | **97.90** ±0.19 | 97.80 ±1.26 |
| InfiniteMNIST | ACC | 97.21 ±0.12 | 97.60 ±0.41 | 99.12 ±0.10 | 95.52 ±0.11 | 98.75 ±0.02 | **99.50** ±0.09 | 95.71 ±0.10 | 99.35 ±0.04 |
| | F-score | 97.21 ±0.14 | 97.60 ±0.41 | 99.12 ±0.10 | 95.51 ±0.11 | 98.75 ±0.02 | **99.50** ±0.08 | 95.71 ±0.10 | 99.34 ±0.04 |
| | Precision | 97.21 ±0.12 | 97.60 ±0.41 | 99.12 ±0.10 | 95.56 ±0.11 | 98.78 ±0.03 | **99.51** ±0.09 | 95.74 ±0.15 | 99.35 ±0.04 |

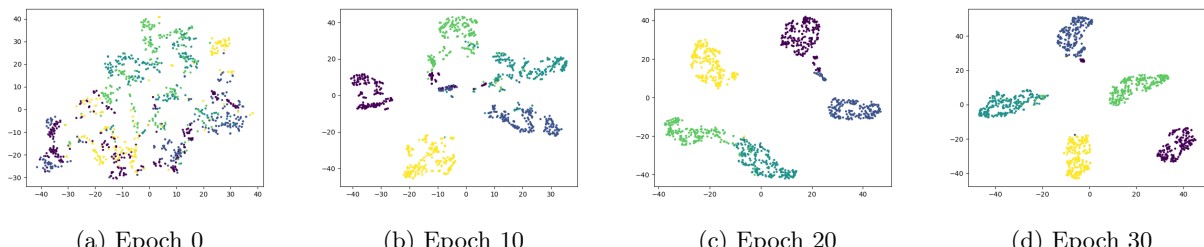

| (a) Epoch 0 | (b) Epoch 10 | (c) Epoch 20 | (d) Epoch 30 |
|---|---|---|---|

Figure 3: Visualization of the unified representation $Y$ during training on the BDGP dataset.

Table 2 presents the classification results across the five datasets. As expected, all methods perform better in classification compared to clustering, since classification benefits from supervised training with label information, whereas clustering relies on uncovering the underlying data structure without supervision, making it inherently more challenging. Notably, SpecRaGE consistently achieves top performance in most datasets and evaluation metrics, with particularly strong results on more complex datasets like Caltech20 and Reuters. Similarly to the clustering results, on InfiniteMNIST, SpecRaGE performs competitively but slightly below the best baseline, suggesting that for simpler, well-structured datasets, multiple methods can achieve near-optimal performance.

**Visualization.** To further demonstrate the effectiveness of the learned unified representation, we utilize the *t*-SNE algorithm on the representation obtained from the validation set during the training. As depicted in Fig. 3, our method successfully separates the data into distinct clusters with increasing training epochs.

### 5.3 Evaluation of Robustness to Data Contamination

In real-world scenarios, data contamination is a pervasive challenge, whether in the form of noise or anomalous outliers, often introduced by faulty sensors, human errors, or external environmental factors. These types of contamination can severely degrade the performance of multi-view models, making robustness a critical capability. To evaluate SpecRaGE's robustness we designed two contamination experiments: one targeting robustness to outliers and the other to Gaussian noise.

In the outlier experiment, global anomalies were randomly injected into one view, following the approach from (Han et al., 2022). For the noise experiment, Gaussian noise with $\sigma = 1.2$ was injected into a portion of the samples in one randomly selected view. We conducted both experiments at contamination ratios of 10%, 20%, 30%, and 40%, and measured clustering ACC.

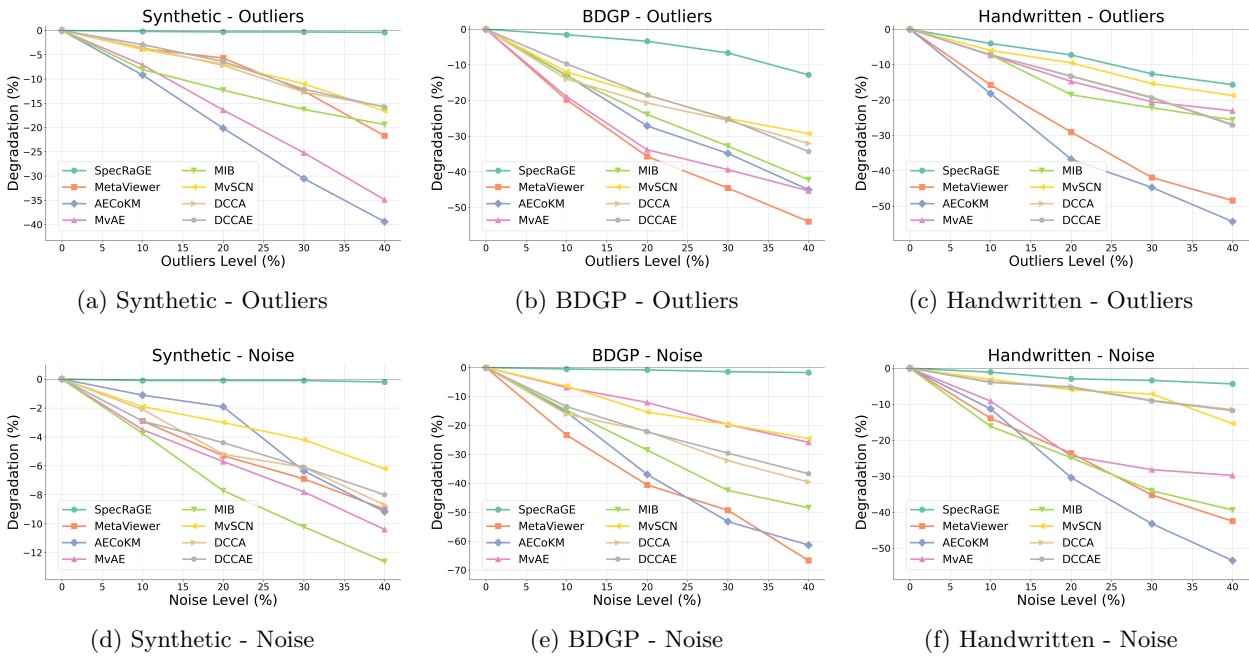

Figure 4: Clustering performance (ACC) degradation across various datasets under increasing levels of outliers (top) and noise (bottom). Each curve represents a different method, illustrating the relative performance drop (%) as the proportion of outliers or noise rises

These tests were performed on two real-world datasets and a synthetic 2-view version of the scikit-learn Blobs dataset. We chose to include this simple synthetic 2D data in this experiment, as it allows us to more clearly distinguish the effect of the noise or outliers on the model's performance. In these experiments, we report the relative degradation percentage with respect to the uncontaminated baseline, where no contamination is injected.

As shown in Fig 4, SpecRaGE consistently outperforms other methods in terms of robustness, maintaining stable performance even under high levels of contamination. In both the outlier and noisy view experiments, SpecRaGE demonstrates significantly smaller relative degradation percentages across all datasets, even at the highest contamination ratios. This superior robustness can be attributed to our fusion-based approach, which fundamentally differs from methods like MvSCN, AECoKM, and MetaViewer that rely on alignment or contrastive objectives. These alignment-based methods attempt to enforce consistency between view-specific representations, which becomes problematic when one view is contaminated, as they may erroneously align clean data with corrupted information. In contrast, SpecRaGE's fusion module can dynamically down-weight contaminated views, preventing their corruption from propagating to the final representation.

## 5.4 Comparison with other Fusion Methods

To evaluate the effectiveness and the relevance of our dynamic fusion module, we conducted a comparison with several popular fusion techniques (Li et al., 2018): Simple Average, Concatenation, Linear layer, and Cross-view Attention (Wu et al., 2022). Among these, the latter two are also learnable fusion approaches. The evaluation was performed with contamination ratios of 10%, 20%, 30%, and 40%, considering both outliers and noise corruption scenarios.

Similar to our robustness analysis in Section 5.3, we measured the relative degradation in clustering performance with respect to an uncontaminated baseline. Figure 5 presents the results of this comparison on the BDGP dataset. In the outlier scenario (Figure 5a), our weighting fusion module demonstrates superior robustness, with performance degrading by only 12% at 40% contamination, while other methods show higher degradation (27.3% for Concat, 23.9% for Simple Average).

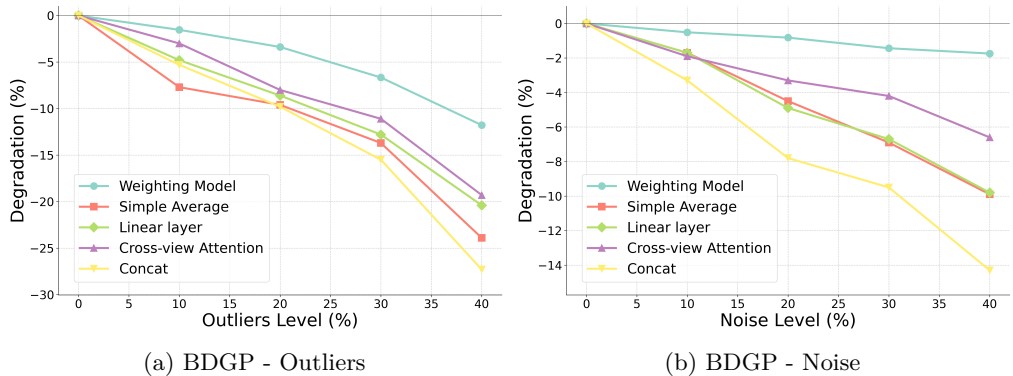

(a) BDGP - Outliers        (b) BDGP - Noise

Figure 5: Comparison of clustering performance (ACC) degradation among different fusion methods on the BDGP dataset with increasing outlier and noise ratios.

When examining noise corruption (Figure 5b), all methods show better resilience compared to the outlier scenario, but still, our fusion maintains remarkably stable performance with only 1.9% degradation at 40% noise, while other methods show higher vulnerability (14.3% for Concat, 9.9% for Simple Average).

These results highlight the relevance and necessity of our fusion module for maintaining robustness across various contamination scenarios.

## 6 Conclusion

In this work, we introduced SpecRaGE, a novel fusion-based framework for multi-view spectral representation learning that effectively integrates graph Laplacian methods with deep learning techniques. By efficiently learning a parametric map to uncover the approximate joint eigenvectors from diverse graph Laplacians, SpecRaGE addresses the challenges of generalizability and scalability in graph Laplacian methods, enabling it to handle large datasets while generalizing to new samples. Moreover, SpecRaGE employs a dynamic fusion technique that enhances robustness against outliers and noise in contaminated multi-view data. Extensive experiments validate that SpecRaGE achieves state-of-the-art performance on standard multi-view benchmarks and significantly outperforms existing methods when faced with outliers and corrupted views. These results highlight SpecRaGE's potential to transform multi-view learning in practical applications where data quality is often compromised.

**Reproducibility Statement.** A complete explanation of SpecRaGE's logic is provided in Section 4 and Algorithm 1. All key details for reproducibility, including the training methodology with the orthogonality constraint, network backbones for both the view-specific and weighting fusion networks, hyperparameters, data splits, operating system, and hardware specifications, are available in Appendix F. Further discussion on the construction of affinity matrices is found in Appendix 4.4.

**Limitations.** While our approach enables scalable and generalizable joint eigenvector approximation, it relies on sufficiently large mini-batches for effective convergence. As discussed in Appendix F, small mini-batches that fail to adequately represent the data distribution can hinder the orthogonalization process, limiting generalization without explicit QR decomposition. Additionally, we observed some sensitivity to high learning rates during the orthogonalization step. Excessively high learning rates can lead to network outputs that contain nearly linearly dependent vectors, causing the QR decomposition to become numerically unstable or fail. To mitigate these limitations, a relatively low learning rate and large batch size are required to ensure stable training.

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

## A Datasets Characteristics

In Table 3, we provide additional information regarding the sample size, number of views, and dimensions of the various datasets.

Table 3: Characteristics of the datasets in our experiments

| Dataset | #Samples | #Classes | #Views | #Dimensions |
|---|---|---|---|---|
| BDGP | 2500 | 5 | 2 | 1750; 79 |
| Reuters | 18,758 | 6 | 5 | 21531; 24892; 34251; 15506; 11547 |
| Caltech20 | 2386 | 20 | 6 | 48; 40; 254; 1984; 512; 928 |
| Handwritten | 2000 | 10 | 2 | 240; 216 |
| InfiniteMNIST | 1,000,000 | 10 | 2 | 784;784 |

## B Proof of Proposition 1

**Proposition 1.** Let $L^{(1)}, L^{(2)}, \ldots, L^{(V)}$ be $n \times n$ real, symmetric, and pairwise commuting matrices. Let $\bar{L} = \sum_{v=1}^{V} L^{(v)}$ be a sum of these matrices. Then, the joint eigenvectors of $L^{(1)}, \ldots, L^{(V)}$ are also the eigenvectors of $\bar{L}$.

*Proof.* Since $L^{(1)}, \ldots, L^{(V)}$ are real, symmetric, and pairwise commuting, they can be simultaneously diagonalized by an orthogonal matrix $U$. This means there exists an orthogonal matrix $U$ such that:

$$U^\top L^{(v)} U = \Lambda^{(v)}, \quad \text{for } v = 1, \ldots, V,$$

where $\Lambda^{(v)}$ are diagonal matrices containing the eigenvalues of $L^{(v)}$, and the columns of $U$ are the joint eigenvectors of $L^{(1)}, \ldots, L^{(V)}$.

Now, consider the sum $\bar{L} = \sum_{v=1}^{V} L^{(v)}$. Using the property of linearity and the simultaneous diagonalization, we have:

$$U^\top \bar{L} U = U^\top \left( \sum_{v=1}^{V} L^{(v)} \right) U = \sum_{v=1}^{V} (U^\top L^{(v)} U) = \sum_{v=1}^{V} \Lambda^{(v)}.$$

Since the sum of diagonal matrices is also a diagonal matrix, $U^\top \bar{L} U$ is diagonal. This implies that $U$ diagonalizes $\bar{L}$. Thus, the columns of $U$, which are the joint eigenvectors of $L^{(1)}, \ldots, L^{(V)}$, are also eigenvectors of $\bar{L}$. $\qquad\square$

## C Equivalence of the loss functions

In Section 4.2, we claim that the loss function in Eq. 2 is equivalent to the joint-diagonalization loss in Eq. 3, that is:

$$\sum_{v=1}^{V} \sum_{i,j=1}^{m} W_{i,j}^{(v)} \|y_i - y_j\|_2^2 = 2 \operatorname{Tr} \left( Y^\top \sum_{v=1}^{V} L^{(v)} Y \right)$$

Here, we provide a proof for this equation.

*Proof.* As demonstrated in (Belkin & Niyogi, 2003) the following relation holds:

$$\sum_{i,j=1}^{m} W_{i,j} \|y_i - y_j\|_2^2 = 2 \operatorname{Tr} \left( Y^\top L Y \right)$$

Therefore, we can derive the following expression:

$$\sum_{v=1}^{v} \sum_{i,j=1}^{m} W_{i,j}^{(v)} \|y_i - y_j\|_2^2 = 2 \sum_{v=1}^{v} \operatorname{Tr} \left( Y^\top L^{(v)} Y \right)$$

$$= 2 \operatorname{Tr} \left( \sum_{v=1}^{v} Y^\top L^{(v)} Y \right) = 2 \operatorname{Tr} \left( Y^\top \sum_{v=1}^{v} L^{(v)} Y \right)$$

$$\square$$

Table 4: Comparison of clustering performance of SpecRaGE against two additional baselines: concatenated Siamese embeddings (Siamese) and spectral embedding on the average Laplacian (SE on $\bar{L}$)

| Datasets | Metrics | Siamese | SE on $\bar{L}$ | SpecRaGE |
|---|---|---|---|---|
| BDGP | ACC | 88.20 | 95.81 | **97.62** |
| | NMI | 75.42 | 91.54 | **93.43** |
| | ARI | 72.21 | 91.79 | **94.10** |
| Reuters | ACC | 46.90 | 53.40 | **56.74** |
| | NMI | 29.61 | 34.43 | **38.31** |
| | ARI | 24.32 | 24.58 | **29.63** |
| Caltech20 | ACC | 38.91 | 46.60 | **50.11** |
| | NMI | 61.48 | 65.37 | **66.02** |
| | ARI | 30.10 | 38.02 | **40.09** |
| Handwritten | ACC | 87.60 | 88.91 | **91.89** |
| | NMI | 82.31 | 84.30 | **86.54** |
| | ARI | 78.89 | 80.92 | **83.02** |
| InfiniteMNIST | ACC | 82.41 | 98.13 | **99.09** |
| | NMI | 85.28 | 96.75 | **97.49** |
| | ARI | 78.62 | 97.23 | **97.90** |

# D  Further experiments

## D.1  Additional Baselines

To further validate our approach, we introduce two additional baselines that help contextualize the performance of SpecRaGE. The first baseline directly concatenates the embeddings obtained from our pre-trained siamese networks across views. This simple approach helps us evaluate whether the additional complexity of our framework is beneficial compared to simply combining the view-specific representations learned by the siamese networks.

The second baseline computes the spectral embedding directly on the average Laplacian $\bar{L}$ using the same affinity matrices obtained from our pre-trained siamese networks. This baseline represents the direct, non-parametric solution to the problem that SpecRaGE aims to solve in a learnable, generalizable way. For fair comparison, as in SpecRaGE, we first construct view-specific similarity matrices using the siamese networks, compute their corresponding Laplacians, average them to obtain $\bar{L}$, and then compute its first $k$ eigenvectors 10 times using the LOBPCG algorithm on different test sets. While this approach provides excellent results as it computes the exact solution, it lacks the ability to generalize to new samples and scale to large datasets - key advantages that our parametric approach provides.

Table 4 presents the clustering performance comparison across all datasets. The concatenated siamese embeddings consistently underperform both SpecRaGE and SE on $\bar{L}$, demonstrating the value of our spectral embedding approach. As expected, SE on $\bar{L}$ achieves strong results across all datasets, as it directly computes the exact eigenvectors that our method aims to approximate. Remarkably, SpecRaGE still outperforms SE on $\bar{L}$ across all datasets. This superior performance can be attributed to SpecRaGE's adaptive weighting mechanism, which learns to create more reliable fused representations by automatically adjusting the contribution of each view based on its quality. The consistent improvement over the direct solution demonstrates that our learnable approach not only successfully approximates the optimal solution but can actually enhance it through intelligent fusion, while providing crucial benefits of generalization and scalability. These results validate both the effectiveness of our approach over simple feature concatenation and show that our parametric approximation not only matches but surpasses the direct solution, benefiting from its robust weighting mechanism while offering the additional advantages of generalization to new samples and scalability to larger datasets.

Table 5: Comparison of clustering performance of SpecRaGE with three InfoNCE-based multi-view methods: CoMVC, AECoKM, and MetaViewer.

| Datasets | Metrics | CoMVC | AECoKM | MetaViewer | SpecRaGE |
|----------|---------|-------|--------|------------|----------|
|          | ACC     | 73.96 | 76.61  | 90.40 | **97.62** |
| BDGP     | NMI     | 59.10 | 68.70  | 86.32 | **93.43** |
|          | ARI     | 49.63 | 54.71  | 89.33 | **94.10** |

### D.2  Comparison with InfoNCE-based Methods

To strengthen our evaluation of SpecRaGE against widely recognized InfoNCE-based multi-view learning methods, we include an additional baseline: CoMVC Trosten et al. (2021), a contrastive learning method that employs an alignment objective to maximize invariance across view-specific representations. In addition to CoMVC, our baseline set already includes AECoKM and MetaViewer, both of which incorporate InfoNCE-style objectives for aligning multi-view representations. Table 5 presents clustering results on the BDGP dataset, showing that our proposed SpecRaGE significantly outperforms all InfoNCE-based baselines, including the newly added CoMVC. This highlights the strength of our method, even in comparison to contrastive learning-based approaches, in learning robust and discriminative multi-view representations.

### D.3  Scalability

As described in Sections 1,2, traditional graph Laplacian-based methods struggle with scalability to large datasets (e.g. Cai et al. (2011); Kumar et al. (2011); Eynard et al. (2012)). In the following experiment, we demonstrate the scalability of our method compared to several well-known traditional approaches in multi-view spectral representation and joint diagonalization of Laplacians. To evaluate scalability, we measure the runtime of each method on the InfiniteMNIST dataset with an increasing number of points. The methods we compare include Jacobi angles for simultaneous diagonalization (JADE) Cardoso & Souloumiac (1996), utilized in Eynard et al. (2012), Co-regularized multi-view spectral clustering (CoRegMVSC) Kumar et al. (2011), and Large-Scale Multi-View Spectral Clustering via Bipartite Graph (LSMVSC) Li et al. (2015). For fairness, the runtime was measured for all methods until convergence (no change in clustering ACC) using an M1 MacBook CPU. For our algorithm, convergence typically occurred after 25 epochs.

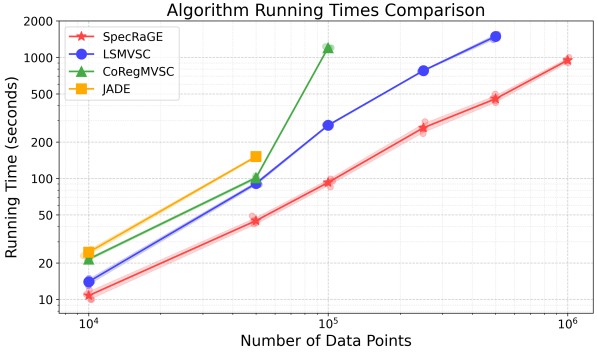

Figure 6: Running time (seconds), as a function of the number of points (dataset size).

As depicted in Fig. 6, SpecRaGE is able to efficiently process large datasets containing up to $1,000,000$ data points within a reasonable timeframe. This scalability is achieved through the implementation of a stochastic training approach on mini-batches. In contrast, JADE terminates after $50,000$ points, CoRegMVSC after $100,000$ points, and LSMVSC after $500,000$ points due to memory constraints, demonstrating the superior scalability of our approach.

### D.4 Approximation of the Joint Eigenvectors

We begin by demonstrating that our unified representation approximates the joint eigenvectors. To show this approximation, we compute the Grassmann distance between the subspace of SpecRaGE's output and that of the true eigenvectors of the matrix $\bar{L} = \sum_{v=1}^{V} L^{(v)}$. These true eigenvectors, referred to as the joint eigenvectors, are computed on a validation set after each training epoch. The squared Grassmann distance measures the sum of squared sines of the angles between two $k$-dimensional subspaces, yielding values within the $[0, k]$ range. To demonstrate this approximation, we used the BDGP dataset and the scikit-learn 2D Blobs dataset, where the second view is generated by applying a random rotation transformation to the first view.

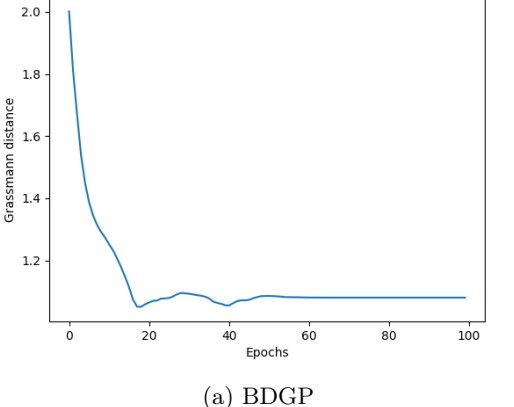 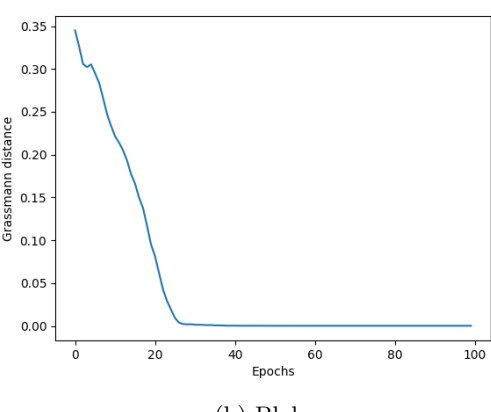

(a) BDGP          (b) Blobs

Figure 7: The Grassmann distance between SpecRaGE's output and the true eigenvectors of the matrix $\bar{L} = \sum_{v=1}^{V} L^{(v)}$ is plotted as a function of the number of epochs. A smaller distance indicates a more accurate approximation of the joint eigenvectors.

In the BDGP dataset, our model has an output dimension of 5, so the maximum distance between the subspace of SpecRaGE's output and that of the joint eigenvectors is also 5. As shown in Fig. 7, this distance significantly decreases early in training and stabilizes around 0.9, indicating that SpecRaGE effectively approximates the joint eigenvectors. For the Blobs dataset, the Grassmann distance quickly approaches zero, indicating that the subspace of SpecRaGE output is extremely close, if not identical, to the subspace spanned by the exact true joint eigenvectors.

In addition, Figure 8 demonstrates that our model learns representations $Y$ that approximately jointly diagonalize the graph Laplacians of both views. Specifically, the matrices $Y^{\top} L^{(v)} Y$ exhibit a predominantly diagonal structure for both the image view (Figure 8a) and the text view (Figure 8b) of the BDGP dataset.

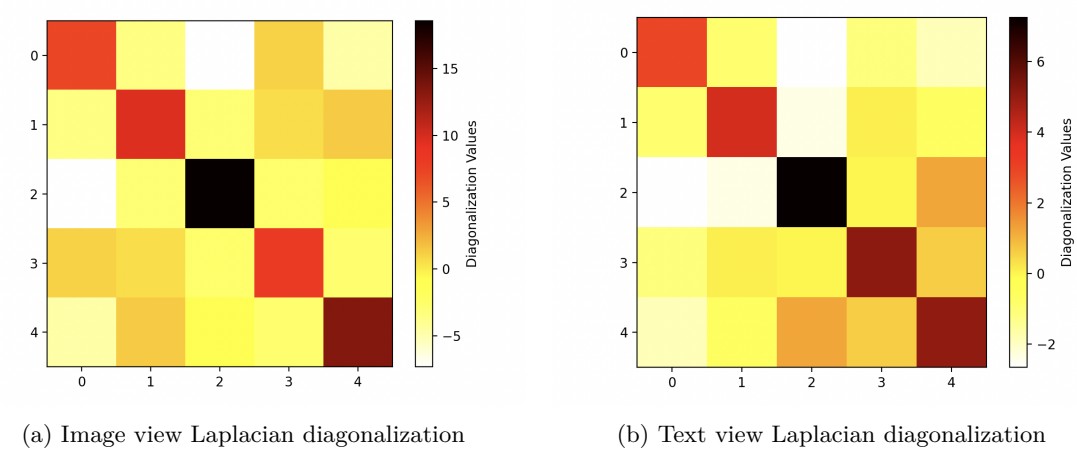

(a) Image view Laplacian diagonalization          (b) Text view Laplacian diagonalization

Figure 8: Approximate joint diagonalization. Visualization of $Y^\top L^{(v)} Y$ for both views of the BDGP dataset, showing the approximate diagonal structure achieved by our learned representations.

## E  Time Complexity Analysis

---
**Algorithm 1** SpecRaGE

---
1:  **Input:** Multi-view data $\hat{x}_1, \hat{x}_2, \ldots, \hat{x}_n$, output dimension $k$, batch size $m$, number of epochs $T$
2:  **Output:** Fused representation $y_1, \ldots, y_n = Y \in \mathbb{R}^{n \times k}$ - the approximate joint eigenvectors.
3:  **for** each epoch $t \in \{1, 2, \ldots, T\}$ **do**
4:      **for** each mini-batch of size $m$ **do**
5:          Obtain view-specific representations $Y^{(1)}, Y^{(2)}, \ldots, Y^{(V)}$
6:          Generate $m$ weights vectors $\alpha_1, \alpha_2, \ldots, \alpha_m$ using the weighting fusion model
7:          Construct $m \times m$ affinity matrices $W^{(1)}, W^{(2)}, \ldots, W^{(V)}$ using Eq. 5
8:          Compute $\tilde{W}_{ij}^{(v)} := W_{ij}^{(v)} \cdot \alpha_i^{(v)} \cdot \alpha_j^{(v)}$ for $v = 1, \ldots, V$
9:          Obtain the fused representation $\tilde{Y}$ using the weights vectors
10:         Apply orthogonality constraint using QR decomposition: $Y = \tilde{Y} R^{-1}$
11:         Compute loss in Eq. 4
12:     **end for**
13: **end for**
14: Forward propagate $\hat{x}_1, \hat{x}_2, \ldots, \hat{x}_n$ and obtain $n$ outputs $y_1, y_2, \ldots, y_n \in \mathbb{R}^k$

---

The time complexity of the algorithm in 1 can be analyzed as follows:

Given that:

- The size of the networks and the number of epochs are constant.

- Batch size is $m$.

- The number of views is $V$.

- The total number of samples is $n$.

- The output dimension is $k$.

The running time breakdown per batch is:

- line 5: $O(mV)$.

- line 6: $O(mV)$.

- lines 7-8: $O(m^2V)$.

- line 9: $O(mV)$.

- line 10: $O(mk^2)$.

- line 12: $O(m^2V)$.

The Overall complexity:

- Per batch: $O(3mV + mk^2 + 2m^2V)$.

- Per epoch: $O(\frac{n}{m} \cdot (3mV + mk^2 + 2m^2V)) = O(n(k^2 + V(2m+3))) = O(n(k^2 + mV))$.

Now since $m$, $V$, and $k$ are much smaller than $n$, this method presents almost linear running time complexity.

## F    Technical Details

For fairness, we run each of the compared algorithms ten times on the above datasets, recording both the mean and standard deviation of their performance. The same backbones are employed across all methods and datasets. Specifically, we used an MLP with hidden layers of sizes 1024, 1024, and 512 for all view-specific networks $g_\theta^{(v)}$ across all datasets and methods. The weighting fusion model also uses an MLP backbone with three hidden layers, each containing 100 units.

Additionally, for certain datasets such as BDGP, InfiniteMNIST, and Reuters, we initially embedded the raw features of each view $\mathcal{X}^{(v)}$ using a pre-trained Autoencoder (AE) with hidden layers of sizes 512, 512, and 2048. This pre-trained AE is used to obtain a lower-dimensional input, typically containing less nuisance information, as shown in (Shaham et al., 2018).

The size of the output of our model ($k$), is determined by the number of categories of the data. In the case of coupled view methods like DCCA, DCCAE, and MIB, we present results based on the two best-performing views. For alignment-based methods, the final unified representations are produced through concatenation. Subsequently, clustering and classification tasks are conducted using $k$-means and SVM classifiers, respectively. Training typically took 35-50 epochs for each dataset.

**Training with Orthogonalization Layer.**    To train the model with the orthogonalization layer, we adopt a technique similar to that used in *SpectralNet* (Shaham et al., 2018). This technique employs a coordinate descent training approach comprising two main optimization steps: the "orthogonalization step" and the "gradient step". Each step involves processing a different mini-batch.

During the orthogonalization step, we forward a mini-batch through the model and compute the QR decomposition of the fused representation to update the weights of the orthogonalization layer. In the gradient step, we pass another mini-batch through the model and use the orthogonalization weights from the preceding orthogonalization step to orthogonal the output. Following this, we compute the loss and update the network weights via backpropagation, while keeping the weights of the orthogonalization layer unchanged.

After training the model, all weights, including those of the orthogonal layer, are fixed. Consistent with observations from *SpectralNet*, we empirically found that when employing large mini-batches, the orthogonalization layer can also approximately orthogonalize the output of other mini-batches towards the end of training. For instance, in the Blobs dataset, when a random batch of size 1024 passes through the model with the fixed orthogonalization layer (i.e., after training), the resulting output $Y$, exhibits approximate orthogonality. This is evident in $Y^\top Y$, where the average deviation of off-diagonal elements from 0 is merely 0.04. In the BDGP dataset, we got an average deviation of ~ 0.05, and in the InfiniteMNIST we got ~ 0.045.

Table 6: Hyper-parameters.

| Hyper-params | BDGP | Reuters | Caltech20 | Handwritten | InfiniteMNIST |
|---|---|---|---|---|---|
| LR | $10^{-3}$ | $10^{-3}$ | $10^{-3}$ | $10^{-3}$ | $10^{-3}$ |
| Batch size | 1024 | 2048 | 1024 | 1024 | 1024 |
| #Neighbors ($l$) | 22 | 18 | 18 | 22 | 30 |
| scale ($\sigma$) | Global; Median | Global; Median | Global; Median | Global; Median | Global; Median |
| Softmax temperture | 250 | 500 | 250 | 250 | 250000 |

**Hyper-parameters.** In Table 6, we provide a breakdown of the hyperparameters utilized for the various datasets. We ensured that the same hyperparameter tuning procedure was applied to all baselines for a fair comparison. Specifically, for each method, we conducted multiple runs (with the number of runs varying based on the method's specific hyperparameters), each using a different hyperparameter configuration. We then selected the setting that yielded the best results across all metrics on a validation set. For the unsupervised case, we used ACC, NMI, and ARI as evaluation metrics, while for classification, we relied on ACC, F-score, and precision.

The #Neighbors parameter corresponds to the number of nearest neighbors ($l$) used for each data point in the Gaussian kernel, as outlined in Appendix 4.4. We select the same number of neighbors for each view $\mathcal{X}^{(v)}$. The scale parameter ($\sigma$) is also utilized for the Gaussian kernel and was chosen for each view $\mathcal{X}^{(v)}$ as the median of the distances from any point in $\mathcal{X}^{(v)}$ to its $l$-nearest neighbors (across all points in the view), resulting in a global scale. The temperature parameter is applied in the softmax function used on the weight vector generated by the weighting fusion model. Using a temperature greater than 1 in the softmax function helps to smooth the weight distribution across the views, reducing the absolute dominance of any single view. This is particularly useful when we want to ensure that all views contribute to the final representation, even if some might be slightly less informative.

The initial learning rate (LR) was uniformly set to $10^{-3}$ for all datasets, with a decay policy in place. This decay policy is contingent on monitoring the validation loss. If the validation loss fails to improve over 10 epochs, the LR is multiplied by 0.1. Furthermore, if the LR decreases to $10^{-8}$, training is terminated. Adam optimizer is used for training.

**Data Split.** For each dataset, we initially divide it into an 80% training set and a 20% testing set. Subsequently, for training, we further divide the training set into a 90% training subset and a 10% validation subset.

**OS and Hardware.** The training procedures were executed on both macOS Sequoia (15.0) and Rocky Linux 9.3, utilizing MacBook M1 processor and Nvidia GPUs including GeForce GTX 1080 Ti and A100 80GB PCIe.

