# OpenReview forum: "Generalizable and Robust Spectral Method for Multi-view Representation Learning"
_TMLR — Accepted by TMLR_

### Review · Reviewer_txCb · 2025-02-10

**Summary Of Contributions:**

The paper presents a way to learn multi-view embedding functions based on the concepts of spectral embedding using the graph Laplacian of different views.  This extends the approach of SpectralMap (Shaham et al., ICLR 2018), which uses a mini-batch formulation of the spectral embedding cost and an optimization that applies a QR decomposition of alternate batches to approximately orthogonalize batch embeddings. That is, a neural network is optimized to provide a mapping that gives coordinates for any input point that act as the coordinates of the $k$ eigenvector associated to the smallest eignenvalues of the Laplacian. The proposed multiview case follows SpectralMap in that the affinity matrix underlying the Laplacians uses a nearest-neighbor truncated heat kernel on top of a representation learned for each view through self-supervised SiameseNets (Koch et al., 2015).  To create robustness on the individual data point the multi-view setting, a multilayer perceptron with softmax activation is applied to each instance to generate weights on the simplex for each view. This enables both weighted averages of the view-specific embeddings to create a unified embedding as well as weighted average of the affinity matrices during learning.

The results use multi view data of varying modality (image, image features, and text for example), quality or difficulty and the proposed outperforms the selected baseline methods. Results also show resilience to the presence of synthetically added noise or outliers to toy and real-world data.

Compared to traditional Laplacian eigenmaps the main contributions are scalability due to mini-batch samples (at the cost of suboptimality), explicit embedding which does not require maintaining entire or quantized representation of training set for out of sample embedding, and a way to downweight outlier views. The first two contributions are shared by SpectralNet, but the last is specific to the multiview case of approximate spectral embedding.

**Audience:**

Yes

**Broader Impact Concerns:**

none.

**Claims And Evidence:**

Yes

**Requested Changes:**

**Fairness of evaluation.**
From the paper, it seems that all methods use the same architecture for the view processing backbone, but there is a subtle concern I have with the results for the proposed method that exploit self-supervised learning for the affinity matrix discussed in Appendix D. Namely, it should be made clear in the main body that this is possibly an information advantage since "The training of the SiameseNets serves as a pre-processing step. Once the SiameseNets are trained, we use them to construct batch affinity matrices for each view during the training."  This means that the proposed work is **distilling** into a multi-view representation the information learned by the SiameseNet. Why not simple learn on top of the representations learned by the SiameseNet to obtain the multi-view embedding? How would a direct concatenation of the SiameseNet outputs across the views perform on the clustering and classification?

In any case, this self-supervised information on what is similar in a view that is used by the proposed method, means the proposed method is actually a two-stage approach. In the first stage contrastive learning could be substituted to the SiameseNet. If this single-view representation is then processed further, say with a linear layer and fine-tuning, my guess is that it would outperform training from scratch. Furthermore, if the other baselines started with this pretraining step per view they may all be more competitive. I understand that the backbone doesn't use SiameseNet directly, but if it did, it and other methods would generally be improved. To be fair, the self-supervised single-view representations could be considered additional views for all methods while learning. This way all methods that are multi-view could benefit from this information. It would also be useful for the community to connect the spectral embedding as a form of multi-view knowledge distillation.

Finally, modern self-supervised learning methods that have been published since SiameseNet, namely SimCLR and CLIP. Perhaps if these were substituted for SiameseNet all methods may benefit more through this pertraining stage. This could be mentioned.

*Adjusting the main body text regarding the pretraining is critical, suggested baseline of concatenated Siamese Net is critical (and should be simple to test). Other additions could strengthen.*

**Broad claims about other methods and contaminated data.**
The paper claims that contrastive learning make assumptions that all views are of similar quality, and "this assumption breaks down in the presence of contaminated data, where enforcing consistency between clear and degraded views can lead to erroneous representations" or later ``Despite their promising performance, these approaches rely on some form of alignment between the view-specific representations, making them vulnerable to data contamination.''. Essentially, this statement claims that contrastive learning are not robust. While the paper's approach of directly learning an instance-wise view weighting is explicitly considering this possibility, there is no ablation study where the 'meta-learning' is removed and SpectralNet is applied to the average affinity. Would the average affinity be more robust? Or is the performance difference due to other differences in methodology?


*An ablation study is necessary to show directly the benefit of 'meta-learning'. Additionally, this ablation should be tested on both the real (no added noise) as well as the synthetically noised data (Figure 4).*

**Cost-benefit of mini-batch.**
Most of the datasets are reasonably small such that standard Laplacian eigenmaps could be applied to the average affinity across views for a baseline. Getting the bottom $k$ eigenvectors of a few thousand instances is not difficult. *The inclusion of the performance of this explicit solution would provide an additional baseline in the paper.*

**Meta-learning and possibility for alternating explicit solution.**
The choice of 'meta-learning' is a bit misleading, as it would initially imply a different or level of learning. How about 'view consistency weights'? A theoretical discussion of Equation 4 is lacking, in terms of the impacts on the optimization or learning. Firstly, while I understand the alpha values are given by an MLP with a softmax layer, it is never very clear that the parameters are part of  $\theta$  in Equation 4. Secondly, if the alpha were additional parameters it appears Equation 4 would be a polynomial of factor 4 in terms of them. However, if the goal was to obtain the spectral embedding of the multiview then only the $\alpha$ in the combined affinity would be needed. That is, it seems completely possible to find a spectral embedding of the reweighted affinity matrix without an explicit network. The idea is to use iterative reweighting where for fixed $\alpha$ weights it is standard Laplacian eigenmaps. Then, to update the weights it would be a linearly constrained (weights for each point on a simplex) quadratic program which could be optimized via Frank-Wolfe. My thought is that this has been done before... In any case, it would shed light on the proposed network solution which can exploit mini-batch.  *It would seem analyzing this, or looking into possibly similar work would be helpful, especially as the `meta-learning' is the main contribution of this work.*


**Hyperparameter selection criteria**
It is critical that the paper is more clear about how hyperparameters were chosen and if similar search was performed on baseline. What is the criterion (performance metric) for hyper-parameter tuning? Is the same criterion used for all tasks (unsupervised clustering and classification decoding)? In the unsupervised case, it is not clear what would be available in the real-world.  Do baseline methods select hyper-parameters, and if not wouldn't it be more fair to use the same criterion?

(One minor note about hyper-parameters, is that the temperature for InfiniteMNIST seems to indicate that all views are given equal weight since this high of temperature surely means a uniform weight... This doesn't seem to make much sense as the second view is just a noisy version. )


**The theoretical analysis is not very applicable**
The theoretical analysis of Proposition 1 is limited to matrices that commute, it does not provide any insight for the approximate case as claimed "we provide additional theoretical insight into the relationship between our loss function in Eq. 3 and the approximate joint diagonalization problem." As stated Corollary 1 is standard Rayleigh-quotient and the eigenvectors of $\bar{L}$ with smallest eigenvalues will be optimal even if the matrices do not commute, which makes it confusing why it is a corollary of Proposition 1. The paragraph concludes with a reference to Appendix E.2 which plots a Grassmanian distance  to the true eigenvectors on the validation sets of two datasets.  For the first dataset the distance converges to a non-zero value, and for the synthetic data 'Blobs' it converges to zero probably should be log scale.
Thus, there is a lack of theoretical analysis to the actual approximation, which isn't required, but *claims should be tempered*.

**Missing baselines**
Trosten et al. (2023) actually proposes multiple methods and the one that is reported to work the best overall is AECoDDC. It is not clear why only one of the methods was compared.

**Clarity on underlying data processing**
 Representation and processing of text Reuters and text in BDGP is missing.

**Synthetic noise**
In the evaluation of Robustness to data contamination, is the noise and anomalies only added to the image view? It is not clear how they would be added in the text view of BDGP dataset.

**Effect of orthogonalization update**
One concern I have stems from the adoption of the orthogonalization algorithm from prior work without inclusion of justification that it is stable and/or optimal. The QR decomposition does not in general give the closest orthogonal representation to the input, so it is not clear how stable this would be across iterations... In the Gram-Schmidt orthogonalization, the first eigenvector which is unmodified in the QR would be stable and ideally this corresponds to the (Fiedler vector), but the subsequent coordinates may note be stable. But the QR in the SpectralNet uses Cholesky decomposition, which may be faster than SVD, but can cause sign flips I believe which would complicate learning batch to batch.

*The paper could benefit form more discussion of the approximate orthogonalization.*

**Blobs**
I am concerned when the paper uses the synthetic Blobs dataset for justifying multiple aspects: the quality of the joint diagonalization, the robustness to contamination, and the approximate orthogonalization... While it is a toy dataset and offer insight, it doesn't offer confirmation that the observations will hold on real data. The paper should be clear about claims only demonstrated on this data.

**Minor points:**

Section 3.1 Minor point is that the affinity matrix needs not be semi-definite and generally won't be if the values between data points that aren't nearest neighbor are set to zero. The resulting unnormalized graph Laplacian will be positive semi-definite for any positive symmetric affinity matrix.


Small error in Appendix D after equation 5 when referring to SiameseNet, it says 'for each view $p$ and then uses $v$ to index view as done elsewhere.

**Strengths And Weaknesses:**

**Strength**
The method as a multiview extension of SpectralNet is straightforward and worthwhile contribution that incorporates the softmax weighting to provide a instance-wise weighting during learning.

The papers is easy to read.

Results seem promising with considerable improvements over baselines.

The work could lead to further developments that incorporate the graph Laplacian formulation as a way to distill multiview similarity information into a single representation. That is, there doesn't seem a reason that the approach couldn't be combined with other self-supervised learning approaches like those based on InfoNCE.

**Weaknesses**
The paper adopts many of the approaches from SpectralNet without mention or discussion in the main body (self-supervised pretraining and orthogonalization update).

A key point in SpectralNet and this paper is that 'The choice of affinity measure plays a crucial role in determining the quality of the generated representations.' However, an explanation that this is based on pertaining with self-supervision is left until Appendix D... It is misleading to not mention this in the main body.

The use of self-supervised pretraining for the affinity matrix, introduces a form of distillation into the multiview learning, which alone is interesting, but it does not seem quite fair without baselines that also use this pretraining step.

The theoretical analysis does not justify approximation.

Hyper-parameter selection is not clear.

Some key points of the data processing and methodology are missing or unclear.

---

> ### Author Response · Authors · 2025-02-23
>
> We wish to thank the reviewer's thoughtful and detailed comments, which have helped us improve the paper. As all weaknesses raised by the reviewer are included in the requested changes, we will address them directly below:
>
> **Requested Changes**:
>
> **Fairness of evaluation**: Thank you for raising this concern. In response to your feedback, we have incorporated a discussion of the pre-training step into the main body of the paper in Section 4.4. Additionally, as suggested, we have introduced the concatenated SiameseNet baseline in Appendix E.1 to assess its performance against our method.
>
> Regarding the fairness of the evaluation, we want to emphasize that while we used pre-trained Siamese networks within our method, the outputs of these networks were not used as direct inputs to our method. Instead, they were solely used to construct the affinity matrices during training (which do not exist in most baselines).
>
> Nonetheless, to further address your concern, we conducted an additional clustering experiment on the BDGP dataset, where we used the Siamese network outputs as inputs for our method and all other baselines. The results, shown in the table below, demonstrate that even when all methods benefit from the same pretraining step, our approach still achieves superior performance.
> | Dataset | Metric | DCCA          | DCCAE         | MvSCN         | MIB           | Multi-VAE     | AECoKM        | MetaViewer                | SpecRaGE                  |
> |---------|--------|--------------|--------------|--------------|--------------|--------------|--------------|--------------------------|--------------------------|
> |    | ACC    | 72.6 ± 1.56  | 74.4 ± 1.40  | 82.4 ± 5.67  | 88.5 ± 2.14  | 51.9 ± 6.66  | 77.3 ± 4.32  | _90.7_ ± 1.28            | **97.3** ± 1.81         |
> |    BDGP            | NMI    | 65.6 ± 1.43  | 70.8 ± 1.20  | 77.3 ± 5.53  | 80.2 ± 2.45  | 27.9 ± 6.11  | 67.4 ± 4.66  | _87.1_ ± 1.22            | **91.8** ± 1.98         |
> |         | ARI    | 49.3 ± 1.22   | 54.8 ± 1.05   | 71.23 ± 6.11  | 75.6 ± 3.01   | 27.1 ± 5.01   | 56.3 ± 4.93   | _89.9_ ± 1.79             | **92.91** ± 1.65          |
>
>
>
> Finally, modern self-supervised methods such as SimCLR and CLIP are designed for specific modalities, making them well-suited for multi-view data involving images, text, or image-text pairs. However, most of the benchmarks we used consist of feature vectors rather than raw images or text, making these methods less directly applicable. Our approach prioritizes a modality-agnostic framework to ensure broad applicability across diverse multi-view scenarios.
>
> **Broad claims about other methods and contaminated data**:  Thank you for this important question about validating the effectiveness of our meta-learning module. We kindly refer the reviewer to Section 5.4, in which we extensively examined the benefit of the meta-learning module by comparing it to various well-known fusion methods. For each fusion method, we ran our method under different levels of noise and outliers ratios, as presented in Figure 5. Second, as the reviewer suggested, we evaluated these fusion methods on clean data with no added contamination in the following table:
> | Dataset | Simple Average | Linear Layer | Attention | Concat  | Meta-Learning (Ours) |
> |---------|---------------|--------------|-----------|---------|----------------------|
> | BDGP    | 96.02 ± 1.10  | _96.92_ ± 1.06 | 94.27 ± 3.43 | 84.68 ± 5.13 | **97.62** ± 0.53  |
>
> Our findings demonstrate that the proposed dynamic fusion mechanism consistently outperforms alternative fusion methods, not only in contaminated scenarios but also in clean scenarios.
>
> Notably, the "Simple average" fusion method serves as the naive, non-weighted counterpart of our approach. As demonstrated in Section 5.4, this method fails to maintain robustness under high noise and outlier ratios, highlighting the importance of our weighting strategy, which effectively downweights corrupted views.
>
> Regarding the possibility of applying SpectralNet directly, we note that SpectralNet operates on single-view data, taking it as input and outputting its spectral embedding. In contrast, our approach extends SpectralNet to the multi-view setting by introducing a fusion mechanism that unifies view-specific representations. Consequently, running SpectralNet directly on multi-view data without fusion is not straightforward.

---

> ### Author Response · Authors · 2025-02-23
>
> **Cost-benefit of mini-batch**: As requested, we have added this baseline to our additional experiments in Appendix E.1, computing spectral embedding directly on the average Laplacian ($\bar{L}$) across views. As you correctly noted, this represents the direct, non-parametric solution to the problem that SpecRaGE aims to solve in a learnable way. As expected, this exact solution achieves strong results across all datasets. Remarkably, SpecRaGE outperforms this baseline across all datasets, which can be attributed to our adaptive weighting mechanism that learns to create more reliable fused representations.
>
> While the direct computation of eigenvectors is indeed feasible for these dataset sizes, our approach offers two crucial advantages: (1) the ability to generalize to new multi-view samples and (2) scalability to larger datasets through mini-batch training. These benefits come with no compromise in performance and demonstrate improved results through our fusion mechanism, as evidenced by our experimental results.
>
> Lastly, we examined the robustness of direct computation on the BDGP dataset and found it demonstrates much higher relative performance degradation than our method (see Figure 4). Specifically, direct computation shows 8\% degradation at 10\% noise, 18\% at 20\% noise, 27\% at 30\% noise, and 38\% at 40\% noise.
>
> **Meta-learning and possibility for alternating explicit solution**:  We recognize that our explanation of the meta-learning module could be more precise. The term "meta-learning" is used because we introduce an additional network that learns a separate but related task—determining how to weigh the view-specific representations to obtain a fused representation. However, we accepted your suggestion and changed this term in the paper to "dynamic weighting model". The parameters of this model are indeed part of $\theta$ and are optimized through the learning process. Notably, the $\alpha_i$ vectors are the outputs of this weighting model and are not independent parameters. Their primary role is to perform a weighted aggregation of the view-specific representations, ensuring that the fused representation accounts for the quality of each view.
>
> Regarding Eq. 4 and its impact on optimization, the loss function naturally guides the meta-learning module to learn appropriate weights by modifying the similarity matrices. Specifically, when multiplying $W_{ij}^{(v)}$ with $\alpha_i^{(v)}$ and $\alpha_j^{(v)}$, assigning high weights to corrupted samples would amplify their distorted similarity values $W_{ij}^{(v)}$, leading to an increased loss value. In contrast, assigning them low weights minimizes their influence on the similarity structure. This creates a self-reinforcing mechanism where the model learns to preserve the structure of reliable samples while automatically down-weighting corrupted ones.  With respect to prior work, some methods, such as Li et al 2015 [4], do introduce a parameter $\alpha$ to perform a weighted average of the Laplacians, formulated as $U^\top \left(\sum_{v=1}^{V}{\alpha^{(v)}L^{(v)}}\right)U$, where $\alpha$ is optimized during the learning process. However, this differs significantly from our approach. In our case, the $\alpha$ values are used at the sample level rather than simply averaging the affinities or the Laplacians. Additionally, our method allows weighting new test samples at inference time using the weighting module.
>
> **Hyperparameter selection criteria**: Thank you for this important comment. We ensured that the same hyperparameter tuning procedure was applied to all baselines for a fair comparison. Specifically, for each method, we conducted multiple runs (with the number of runs varying based on the method's specific hyperparameters), each using a different hyperparameter configuration. We then selected the setting that yielded the best results across all metrics on a validation set. For the unsupervised case, we used ACC, NMI, and ARI as evaluation metrics, while for classification, we relied on ACC, F-score, and precision. Regarding the concern about real-world hyperparameter selection in the unsupervised case, we followed a common practice of using a held-out validation set, ensuring that the selected hyperparameters generalize well without relying on ground-truth labels during testing. We updated the paper with these details as well.
>
> Regarding your note about the temperature parameter in InfiniteMNIST. After revisiting this, we conducted additional experiments with lower temperature values (250, 500, and 1000) and found that the results remained consistent. We attribute this to the fact that the second view was generated with relatively small Gaussian noise ($\sigma = 0.2$), preserving a strong correlation with the original view. As a result, even when applying uniform weights (which means a simple averaging of the view-specific representations), the model can still achieve optimal performance when no other contamination is injected.

---

> > ### Comment · Reviewer_txCb · 2025-03-05
> >
> > The revision improves the paper and addresses many of my concerns.
> >
> > In the context of the work by Li et al. [4], how different the dynamic formulation is from the static view weighting?  Empirically, I expect the gap static weighting performance to be between $\bar{L}$ and the dynamic weighting. The level of variance of the dynamic weighting could be assessed roughly looking at the average and standard deviation of weights across instances, to see if the same view is down weighted across samples.
> >
> > A minor nitpick is that there is a missing comma in a a series following an ellipse $x_1,\ldots x_n$-> $x_1,\ldots, x_n$.

---

> > > ### Author Response · Authors · 2025-03-07
> > >
> > > Thanks for the response.
> > > We conducted an additional experiment on the BDGP dataset to directly compare sample-specific weighting with static view weighting. Following your suggestion, we implemented a static weighting approach where we calculated the average weights across the entire batch and applied these average values uniformly to all samples during fusion.
> > > With this static weighting approach, we observed performance degradation of 1.5% with 10% noise, 2.2% with 20% noise, 4.0% with 30% noise, and 6.5% with 40% noise. Comparing to the results in Figure 5, this performance degradation was approximately 4% steeper than our sample-specific approach but less severe than simple averaging without weighting, which aligns with your expectation that static weighting would perform between these two approaches.
> > > To further investigate the variance in weights, we examined the distribution of weights for samples with 20% noise injection in the first view on BDGP. The mean weight for the first view was 0.35 with a standard deviation of 0.09, indicating meaningful variation in sample-specific weighting. In Figure 2, we also show the distribution of the weights between clean and contaminated samples.

---

> ### Author Response · Authors · 2025-02-23
>
> **The theoretical analysis is not very applicable**: Thank you for this insightful comment. Our theoretical analysis indeed assumes the case where all Laplacians commute, primarily to establish a clear connection between our loss function and the joint diagonalization problem. However, prior works [1, 2] have extensively discussed that this loss remains a reasonable approximation even when the Laplacians do not necessarily commute.
>
> Additionally, we have expanded Appendix E.2 to include two new plots on the BDGP dataset, demonstrating that the output representations $Y$ approximately diagonalize each Laplacian. Specifically, we showed that $Y^\top L^{(v)}Y$ is approximately diagonal.
>
> Regarding Corollary 1, our goal was to highlight that the minimizer of our loss (which is the eigenvectors of the average Laplacian $\bar{L}$) also serve as the **joint** eigenvectors of $\( L^{(1)}, L^{(2)}, \dots, L^{(V)} \)$. We have refined the statement in Corollary 1 to make it clearer.
>
> **Missing baselines**: Thank you for your suggestion. While Trosten et al. propose multiple methods, their work primarily focuses on deep clustering rather than representation learning. We specifically chose AECoKM because it aligns with our two-stage evaluation framework, where representation learning and clustering are separate processes. Their other methods, including AECoDDC, are end-to-end deep clustering models that directly predict cluster assignments during training. This makes them less suitable for our study since we focus specifically on learning high-quality representations that can be used for various downstream tasks, not just clustering. Moreover, AECoDDC and AECoKM share the same representation learning architecture, differing only in their clustering approach.
>
> **Clarity on underlying data processing**: Following established practice in prior works, we utilized the Reuters and BDGP datasets in their pre-processed format, which is how they are publicly distributed in the research community. In both datasets, the raw text has already been converted into feature vectors, as detailed in Section 5.1 and Appendix A.
>
> **Synthetic noise**: Since both text and image input views are represented as feature vectors in our experiments, it was possible to inject contamination into both views regardless of their original modality.
>
> **Effect of orthogonalization update**: Thank you for this important comment. The theoretical justification for the stability of our orthogonalization layer can be drawn from Belkin and Niyogi’s work [3] on eigenvector convergence. As mini-batch sizes increase, they provide a more faithful approximation of the underlying data distribution, allowing the parametric mappings introduced in SpectralNet to converge to the eigenfunctions of the Laplace-Beltrami operator. Since these eigenfunctions are uniquely determined by the data manifold’s geometry, the learned representations remain stable across batches.
>
> Regarding your concerns about QR stability and sign flips, our implementation explicitly addresses this issue by normalizing the signs of diagonal elements in R, ensuring consistent sign choices, and avoiding the learning instability that would result from arbitrary sign flips.
>
> To further support the stability of this orthogonalization, it can be observed in Appendix E.2 that the Grassman distance, *which was computed on a validation set every epoch without explicit QR decomposition*, smoothly decreases over time.  Additionally, we've extended the analysis of off-diagonal elements of $Y^\top Y$ to BDGP and NoisyMNIST datasets, showing average deviations of approximately 0.05 and 0.045, respectively, demonstrating that the orthogonalization layer indeed converges consistently.
>
> **Blobs**: We acknowledge that the Blobs dataset is a synthetic benchmark, and we use it primarily as a complementary tool to provide clear, interpretable insights into specific aspects of our method. For instance, in the robustness experiment in Section 5.3, we wanted to demonstrate that even with this simple synthetic dataset, where all methods perform well in the clean scenario, they struggle when faced with data contamination.
>
> **Minor Points**: Thank you for this catch!
> We fixed the error with the $p$ index.
> Regarding Section 3.1, you are correct, and $W$ does not necessarily need to be positive semi-definite. We fixed this accordingly in the paper.
>
> [1] Davide Eynard, Klaus Glashoff, Michael M Bronstein, and Alexander M Bronstein. Multimodal diffusion
> geometry by joint diagonalization of laplacians.  2012.
>
> [2] Davide Eynard, Artiom Kovnatsky, Michael M Bronstein, Klaus Glashoff, and Alexander M Bronstein.
> Multimodal manifold analysis by simultaneous diagonalization of laplacians.  2015
>
> [3] Mikhail Belkin and Partha Niyogi. Convergence of laplacian eigenmaps. 2006
>
> [4] Yeqing Li, Feiping Nie, Heng Huang, and Junzhou Huang. Large-scale multi-view spectral clustering via
> bipartite graph. 2015.

---

### Review · Reviewer_p3ct · 2025-02-14

**Summary Of Contributions:**

This paper presents SpecRaGE, a spectral-based multi-view representation learning (MvRL) framework that integrates graph Laplacians with deep learning to improve robustness. By leveraging approximate joint diagonalization via a parametric neural network, SpecRaGE avoids alignment constraints used in contrastive learning. Additionally, a meta-learning fusion module dynamically reweights views based on data quality, enhancing robustness to noise and outliers. Empirical results demonstrate better performance in clustering and classification, particularly under data contamination.

**Audience:**

Yes

**Claims And Evidence:**

Yes

**Requested Changes:**

1.	Expand the Related Work section to explicitly discuss scalability challenges in Laplacian-based methods and compare with large-scale spectral techniques.
2.	The meta-learning module assigns weights to views based on learned reliability, but what are the learned patterns? How is the loss function for this meta-learning module defined? Does the meta-learning network receive supervision signals directly?
3.	The paper claims that SpecRaGE is effective in handling incomplete views, yet no experiments validate this. Since missing data is fundamentally different from noisy or corrupted data, additional empirical validation is necessary to support this claim.
4.	While each step of SpecRaGE algorithm is well described in Section 4, it would greatly help readers better understand the algorithm if the pseudo-code for SpecRaGE is presented.
5.	Although the paper compares the proposed method with existing approaches, it could be strengthened by including more recent or alternative state-of-the-art methods that also address similar issues presented in paper.

**Strengths And Weaknesses:**

S1.	The introduction of approximate joint diagonalization of graph Laplacian using deep learning is novel and eliminates the need for explicit alignment between views, reducing sensitivity to noise.
S2.	The meta-learning fusion module effectively re-weights unreliable views based on learned data quality, mitigating the negative impact of contaminated views.
S3.	The paper includes a mathematical formulation connecting its loss function to the joint diagonalization problem, reinforcing its conceptual rigor.
S4.	The method is extensively evaluated on five diverse datasets, demonstrating consistent improvements in clustering and classification performance over strong baselines.

W1.	The paper lacks a detailed theoretical justification for choosing Graph Laplacian-based MvRL and its specific formulation, making the methodological foundation less clear.
W2.	The weighting process of the meta-learning fusion module is not well explained, particularly in terms of how it interprets contamination levels across different views.
W3.	The paper could benefit from a more detailed discussion on scenarios where the proposed method may not perform as expected. Understanding the limitations and failure cases can provide more nuanced insights and guide practical implementations.
W4.	The Related Work is not sufficiently comprehensive, as it lacks a detailed discussion on scalability challenges in traditional Laplacian-based methods

---

> ### Author Response · Authors · 2025-02-23
>
> Thank you for reviewing this paper and helping us improve this manuscript. Below our responses for both weaknesses and requested changes:
>
> **W1**. Regarding theoretical justification, in Sections 1 and 2, we extensively discuss the advantages of graph Laplacians in representation learning, primarily due to their eigenvectors' ability to encode both local similarity relationships and global structure while simultaneously enabling dimensionality reduction. Given these advantages, extending graph Laplacian methods to the multi-view setting has been a natural and widely adopted approach.
> Regarding formulation of graph Laplacian-based methods, most multi-view spectral methods rely on a variance of the the following Rayleigh quotient: $\min_{U \in \mathbb{R}^{n \times k}} \text{trace}{\left(U^\top \bar{L}U\right)}, \quad \text{s.t.} \quad U^\top U = I$ where $\bar{L}$ represents a form of average of the graph Laplacians from each view as we discussed in Section 3.2.
>
> **W2**. See our response to "2. Meta-learning module"
>
> **W3**. Regarding limitations, while our approach approximates joint eigenvectors stochastically, it requires sufficiently large mini-batches to converge effectively. As detailed in Appendix F, small batches that don't adequately represent the data distribution can prevent convergence of the orthogonalization process, limiting the method's ability to generalize to new data without explicit QR decomposition. Second, we observed some sensitivity to high learning rates during the orthogonalization step. With high learning rates, the network can produce outputs that cause the QR decomposition to fail numerically. This occurs because aggressive updates can lead to nearly linearly dependent vectors in the network's output, making the QR factorization ill-conditioned or impossible to compute. This limitation necessitates a relatively low learning rate to maintain numerical stability throughout training. We added a limitation paragraph to the paper discussing these limitations.
>
> **W4**. See our responses to "1. Related work" and "5. More recent related methods"
>
> **Requested Changes**:
>
> 1. **Related work**: We have expanded the discussion in the Related Work section to explicitly address scalability challenges in Laplacian-based methods and compare them with large-scale spectral techniques. Please refer to Section 2 under the paragraph "Graph Laplacian-based MvRL Methods."
>
> 2. **Meta-learning module**: The meta-learning module learns to assign weights through direct feedback from our loss function in Eq. (4) without requiring explicit supervision. This loss allows the meta-learning model to dynamically adjust the contribution of each view. By multiplying $W_{ij}^{(v)}$ with $\alpha_i^{(v)}$ and $\alpha_j^{(v)}$, the model effectively reduces the influence of noisy or corrupted samples, ensuring that their similarities to other samples are down-weighted while preserving the structure of reliable ones. If the meta-learning model assigned high weights to contaminated views, the loss would increase, as noisy samples would distort the similarity structure in the corresponding $W^{(v)}$ and lead to suboptimal representations. Overall, this creates a self-reinforcing mechanism where the model learns to identify reliability patterns: high-quality views receive higher weights, while corrupted ones are automatically downweighted. We reshaped Section 4.3 in the paper to make it more clear. Additionally, please note that in the revised version, we have removed the term 'meta-learning' as requested by one of the reviewers. Instead, we now use 'weighting model' or 'weighting fusion model'.
>
> 3. **Incomplete views**: Thank you for this observation. You are absolutely right, and we did not intend to claim that SpecRaGE explicitly handles missing views. Our use of the term "incomplete" referred to anomalous or noisy views rather than entirely missing data. We have revised the paper to use more precise terminology to avoid ambiguity. To further explore the potential of SpecRaGE in handling missing views, we conducted an additional experiment where we randomly zeroed out 10\% and 20\% of the samples in different views in the BDGP dataset. The resulting clustering performance showed relatively mild degradation—2.6\% and 6.5\%, respectively—suggesting that SpecRaGE may also have the capacity to handle missing views. However, since handling missing views is a distinct challenge with its own line of research, we leave a more thorough investigation of this aspect for future work.
>
> 4. **Pseudo code**: We have included the complete pseudo-code of SpecRaGE along with runtime analysis in Appendix E. This can be found specifically in Algorithm 1 of Appendix E for easier reference.

---

> ### Author Response · Authors · 2025-02-23
>
> 5. **More recent related methods**: We have identified several recent methods addressing robustness in multi-view learning. The most relevant are [1], which handles view uncertainty through probabilistic modeling, and [2], which uses modal regression for outlier robustness. However, these methods have limitations - [1] primarily handles Gaussian noise, while [2] requires complex optimization with multiple terms. Additionally, their code implementations were not publicly available for experimental comparison.
> Other recent works like [3] and [4] address different aspects (correspondence noise and label noise, respectively) and are less relevant to our focus on view corruption. We added a detailed discussion of these approaches to our related work section.
>
> [1] Yu Geng, Zongbo Han, Changqing Zhang, and Qinghua Hu. Uncertainty-aware multi-view representation
> learning. In Proceedings of the AAAI Conference on Artificial Intelligence, volume 35, pp. 7545–7553, 2021.
>
> [2] Jiamiao Xu, Fangzhao Wang, Qinmu Peng, Xinge You, Shuo Wang, Xiao-Yuan Jing, and CL Philip Chen.
> Modal-regression-based structured low-rank matrix recovery for multiview learning. IEEE Transactions on
> Neural Networks and Learning Systems, 32(3):1204–1216, 2020.
>
> [3] Yuan Sun, Yang Qin, Yongxiang Li, Dezhong Peng, Xi Peng, and Peng Hu. Robust multi-view clustering
> with noisy correspondence. IEEE Transactions on Knowledge and Data Engineering, 2024.
>
> [4] Cai Xu, Yilin Zhang, Ziyu Guan, and Wei Zhao. Trusted multi-view learning with label noise. arXiv preprint
> arXiv:2404.11944, 2024.

---

### Review · Reviewer_aZ3f · 2025-02-20

**Summary Of Contributions:**

In this paper, authors  introduced  SpecRaGE, a novel fusion-based framework that integrates the strengths of graph Laplacian methods with the power of deep learning to overcome these challenges. SpecRage uses neural networks to learn parametric mapping that approximates a joint diagonalization of graph Laplacians. This solution bypasses the need for alignment while enabling generalizable and scalable learning of informative and meaningful representations. Moreover, it incorporates a meta-learning fusion module that dynamically adapts to data quality, ensuring robustness against outliers and noisy views. Extensive experiments demonstrate that SpecRaGE outperforms state-of-the-art methods, particularly in scenarios with data contamination, paving the way for more reliable and efficient multi-view learning.

**Audience:**

Yes

**Broader Impact Concerns:**

NAN

**Claims And Evidence:**

Yes

**Requested Changes:**

NaN

**Strengths And Weaknesses:**

(+)This is a very interesting job, and in this job, contribution is enough. The writing and organization of the paper are very well-organized.

There are some minor issues that require the author's explanation and supplementation.

(-) It is suggested that the author discuss the limitations of some methods.

(-) It is recommended that the author show or compare the training time, testing time, and number of parameters with the current well-known methods.

---

> ### Author Response · Authors · 2025-02-23
>
> We sincerely appreciate the reviewer's comments and are glad they find this work interesting. Below are our responses:
>
> **Limitations**: Regarding the limitations of our method, we added to the paper a limitation paragraph discussing certain limitations of this approach. Specifically, SpecRaGE requires sufficiently large mini-batches to converge effectively. As detailed in Appendix F, small batches that don't adequately represent the data distribution can prevent convergence of the orthogonalization process, limiting the method's ability to generalize to new data without explicit QR decomposition. Second, we observed some sensitivity to high learning rates during the orthogonalization step. With high learning rates, the network can produce outputs that cause the QR decomposition to fail numerically. This occurs because aggressive updates can lead to nearly linearly dependent vectors in the network's output, making the QR factorization ill-conditioned or impossible to compute. This limitation necessitates a relatively low learning rate to maintain numerical stability throughout training. Regarding the limitations of recent methods in this field, we kindly refer the reviewer to Section 2 where we extensively discussed recent and related works and their limitations in the context of our paper.
>
> **Running time**: As discussed in the paper, SpecRaGE is designed to be a scalable graph Laplacian-based approach compared to traditional methods. To empirically validate this, we conducted an experiment in Appendix D.2, demonstrating that SpecRaGE processes large datasets efficiently, maintaining a reasonable training time compared to other graph Laplacian-based methods. Additionally, Appendix F provides a detailed breakdown of the model architectures, including the number of layers and layer sizes of SpecRaGE and the baselines.

---

> ### Comment · Reviewer_aZ3f · 2025-04-03
> **I  agree to accept this work.**
>
> Thank you very much for your response. The concerns I had have been fully addressed. I  agree to accept this work.

---

### Decision · Action_Editor_ySy2 · 2025-04-06

**Recommendation:** Accept with minor revision

**Comment:**

This work introduces a framework called SpecRaGE, which integrates the strengths of graph Laplacian methods with the power of deep learning to address the reliability and efficiency challenges in multi-view representation learning. Three reviewers have provided positive feedback regarding its novelty, contributions, mathematical rigor, extensive evaluation, readability, and significant improvements over baseline methods. However, they also raised several concerns, including the lack of detailed justification, explanation, or discussion, the comprehensiveness of the related work, the extensiveness and fairness of the experimental study, and the need for an additional ablation study to better support the role of meta-learning in this work, among others. The authors have provided thoughtful responses to address each of these concerns and have revised the work accordingly. Reviewer txCb, who provided the most detailed review, commented that "the revision improves the paper and addresses many of my concerns."

In their official recommendation, all three reviewers recommended "Leaning Accept," which are consistent with each other. Additionally, Reviewer txCb suggested that more comparisons could be made with respect to InfoNCE-based multi-view learning methods, and that the theoretical analysis could be further refined to directly address the approximation.

It is evident that certain members of TMLR's audience would find the findings of this paper valuable. Furthermore, the claims made in the submission are overall sufficiently supported by clear, convincing, and accurate evidence. Considering the reviews, the authors' responses, and the final recommendations, the AE recommends "Accept with minor revisions." This will also allow the authors to address the above comments raised by Reviewer txCb.

**Audience:**

Yes

**Claims And Evidence:**

Yes

---

> ### Author Response · Authors · 2025-04-17
> **Author Response to Minor Revisions**
>
> Thank you for your efforts in managing the review process of our paper.
>
> In response to the reviewer’s suggestion to include further comparisons with InfoNCE-based multi-view learning methods, we have added a new evaluation in Appendix D.2, incorporating an additional baseline (CoMVC) alongside the existing ones.
>
> Additionally, regarding the theoretical analysis, we have already included a new paragraph in Section 4.5 to clarify that exact joint diagonalization is uncommon in real-world data, and our objective aims to identify approximate joint eigenvectors instead. We support this claim empirically in Appendix D.4, where we present two experiments demonstrating that our method effectively approximates the true eigenvectors.